# Estimating Grassland Curing with Remotely Sensed Data

Wasin Chaivaranont[1], Jason P. Evans[1], Yi Y. Liu[1, 2], Jason J. Sharples[3]

[1]ARC Centre of Excellence for Climate Systems Science and Climate Change Research Centre, UNSW, Sydney, NSW 2052, Australia

[2]School of Geographical Sciences, Nanjing University of Information Science and Technology, Nanjing, 210044, China

[3]School of Physical, Environmental and Mathematical Sciences, UNSW, Canberra, ACT 2600

*Correspondence to:* Wasin Chaivaranont (w.chaivaranont@student.unsw.edu.au)

**Abstract.** Wildfire can become a catastrophic natural hazard, especially during dry summer seasons in Australia. Severity is influenced by various meteorological, geographical, and fuel characteristics. Modified Mark 4 McArthur's Grassland Fire Danger Index (GFDI) is a commonly used approach to determine the fire danger level in grassland ecosystems. The degree of curing (DOC, i.e. proportion of dead material) of the grass is one key ingredient in determining the fire danger. It is difficult to collect accurate DOC information in the field, therefore, ground observed measurements are rather limited. In this study, we explore the possibility of whether adding satellite observed data responding to vegetation water content (Vegetation Optical Depth, VOD) will improve DOC prediction when compared with the existing satellite observed data responding to vegetation greenness (Normalised Difference Vegetation Index, NDVI) based DOC prediction models. First, statistically significant relationships are established between selected ground observed DOC and satellite observed vegetation datasets (NDVI and VOD) with an $r^2$ up to 0.67. DOC levels estimated using satellite observations were then evaluated using field measurements with an $r^2$ of 0.44 to 0.55. Results suggest that VOD based DOC estimation can reasonably reproduce ground based observations in space and time and is comparable to the existing NDVI based DOC estimation models.

**Copyright statement**

## 1. Introduction

Wildfire can be responsible for major environmental damage or changes to ecosystems (Cobb et al., 2016; Gazzard et al., 2016; Mistry et al., 2016). One of the important components in determining the severity of wildfire is fuel availability. Wildland fuels can vary considerably, both spatially and temporally (Stambaugh et al., 2011). Various interpretations and characterisations of fuel have been made in past studies as a key contribution to assessing wildfire potential (Hudec and Peterson, 2012; Jurdao et al., 2012; Sharples et al., 2009b; Stambaugh et al., 2011; Yebra et al., 2013). Fuel can also be quantified by its age or time since last fire (Bradstock et al., 2010).

In this study, we focus on the availability of combustible fuel in the aboveground biomass in grassland ecosystems; this fuel availability metric is referred to as the degree of curing (DOC). The DOC is the percentage of dead material in a grassland fuel bed; 100 % indicates a fully cured (dead) grassland fuel complex. The DOC has a direct influence on wildfire development in grasslands, hence, it is an important input for fire danger indices and fire spread models, such as the McArthur Grassland Fire Danger Index (GFDI) (Gill et al., 2010) and the CSIRO grassland fire spread model (Cruz et al., 2015; Kidnie et al., 2015).

Generally, fires are unable to spread across grasslands that are less than 50 % cured (Anderson et al., 2011). Though this lower limit has been revised, since a more recent study demonstrated that fire can spread in grassland with DOC as low as 20 % (Cruz et al., 2015). In climatological studies, DOC is often assumed to be 100 % (Pitman et al., 2007). This leads to an overestimation of areas experiencing high levels of fire danger and hence provides only a weak indication of where to focus

resources of fire agencies. An accurate, spatially and temporally explicit, estimate of DOC would provide more useful guidance to these agencies.

Measuring DOC in the field is a tedious and expensive task, especially when an accurate assessment of curing is required. Anderson et al. (2011) suggested that current methods for measuring DOC still present problems. The visual assessment method, which relies on field observers to estimate the general curing value based on their expertise and a visual guide, is subjective and can often be unrepresentative of DOC of the entire area. Destructive sampling approaches can provide accurate field based observation, but is a labour intensive task. Thus, Anderson et al. (2011) offered a simple field based method utilising a levy rod, based on the modified point quadrant method of pasture assessment; the approach involves counting the number of live and dead touches on a thin steel rod that was driven into the ground. It was suggested that this approach can be applied across Australia with higher accuracy than current visual assessment methods (Anderson et al., 2011).

Apart from ground measurement, DOC can be estimated using satellite remotely-sensed data, but it is limited by the satellite sensors' capability, e.g. spatial resolution and various atmospheric interferences. Dilley et al. (2004) established a relationship between curing and Normalised Difference Vegetation Index (NDVI, a proxy of vegetation canopy greenness) by estimating live fuel moisture content from NDVI and relating it to curing via an exponential function using a finite difference Levenberg— Marquardt method (Dilley et al., 2004; Rouse et al., 1973). Newnham et al. (2011) showed that estimation of curing using a relative greenness (RG) approach that was based on NDVI distribution provided more accurate estimation of curing than a direct linear regression between curing and NDVI. (Chladil and Nunez, 1995) used curing derived from a soil dryness index model and NDVI to predict soil and fuel moisture content. Various optical based vegetation indices computed from remote sensing reflectance products can also be developed into a satellite based model integrated with ground observations to predict curing (Martin et al., 2015; Turner et al., 2011). These methods, though vastly different in their approaches, achieved good results for their set objectives, but they tend to focus on particular applications. It should also be noted that optical based remote sensing products, including NDVI, are affected by cloud cover and aerosols. Some studies explicitly acknowledge challenges presented by cloud effects and when there are both forest and water bodies in the same NDVI pixel, which results in an erroneous grassland interpretation (Allan et al., 2003; Chladil and Nunez, 1995). However, if appropriately detailed aerosol data is available, atmospheric correction can mitigate the aerosol effect on NDVI.

Currently, there are satellite based DOC products over Australia provided by Bureau of Meteorology. The products have 500 m spatial resolution and 8 day temporal resolution, and are based on two past studies (Martin et al., 2015; Newnham et al., 2010). There are five separate satellite based DOC models; four are from Newnham et al. (2010) and one is from Martin et al. (2015). All satellite based DOC models here are based on optical and near-infrared wavelength bands. We would like to investigate whether including a recent passive microwave based satellite product can improve the DOC estimation over Australia or not.

A passive microwave based remote sensing vegetation product, referred to as Vegetation Optical Depth (VOD), has been developed recently (Meesters et al., 2005). VOD is primarily sensitive to vegetation water content, including both leafy and woody components (Guglielmetti et al., 2007; Jackson and Schmugge, 1991; Kerr and Njoku, 1990). Unlike the traditional optical based vegetation indices, such as NDVI, VOD is minimally influenced by the atmospheric conditions due to its longer wavelength and stronger penetration capacity (Jones et al., 2009). However, it has a coarser spatial resolution (0.1°) in comparison with optical based products, which is a consequence of the low energy microwave emissions from the Earth's surface. It has been demonstrated that VOD can capture the changes in vegetation water content over different land cover types at the global scale, including grassland, cropland, savannas, tropical forests, and boreal forests (Liu et al., 2013a, 2013b, 2015). A recent study on estimating fire severity based on VOD changes also demonstrated that VOD has high correlation with aboveground biomass in Australian tropical savannahs that includes large area of northern grassland with an $r^2$ of 0.96 (Chen et al., 2018). Also, NDVI and VOD provide complementary information and can comprehensively characterise vegetation dynamics when combined (Andela et al., 2013).

There are two objectives of this study. The first is to explore the possibility of whether adding the VOD (responding to vegetation water content) will improve DOC prediction when compared with existing NDVI (responding to vegetation greenness) based DOC prediction model. The second is to implement the satellite based DOC estimation (both our and existing models) into GFDI to investigate whether better fire severity predictions can be achieved in grassland environments.

## 2. Materials

### 2.1 Satellite based Products

The NDVI dataset used here is derived from the Moderate Resolution Imaging Spectroradiometer (MODIS) 8 day surface reflectance product (MOD09A1) on-board the Terra satellite (Vermote and Vermeulen, 1999). The MOD09A1 product used here for computing NDVI is an 8 day product, which has less noise than the daily product, and its spatial resolution is 0.005° (~500 m). This product is obtained from Remote Sensing at National Computational Infrastructure (NCI) MODIS Land Product for Australia website (Paget and King, 2008). It is produced from original tiles provide by the United States Geological Survey (USGS) for Australia with a starting date from February 18th, 2000.

The 8 day NDVI data is derived from the MODIS reflectance dataset using the following Eq. (1):

$$\text{NDVI} = \frac{\rho_2 - \rho_1}{\rho_2 + \rho_1} \tag{1}$$

where $\rho_1$ and $\rho_2$ are spectral reflectance measurements obtained from the visible (red) and near-infrared regions, respectively. During the conversion, to ensure the quality of data, only pixels with ideal quality in all bands and a view angle zenith of less than 60° were kept for the analysis, as suggested by Newnham et al. (2011). The spatial resolution is kept at 0.005°.

The VOD dataset used here is retrieved from the Advanced Microwave Scanning Radiometer – Earth Observing System (AMSR-E) and derived using the Land Parameter Retrieval Model (LPRM) approach from which soil moisture and VOD are retrieved simultaneously (Meesters et al., 2005; Owe et al., 2001). Several assumptions are made in the LPRM approach, including: canopy surface temperature equal to soil surface temperature, a constant single scattering albedo, same vegetation parameters for both Horizontal and Vertical polarizations, and minimal effect of surface roughness (Meesters et al., 2005; Owe et al., 2001). Uncertainties in soil moisture and VOD retrievals are expected with these assumptions. The evaluation of LPRM soil moisture over Australia showed that the temporal patterns of satellite-based and in situ soil moisture agree very well (Draper et al., 2009; Gevaert et al., 2016). This agreement suggests a reasonable separation of temporal patterns of soil moisture and VOD, while uncertainties may exist in the absolute magnitudes of these two variables.

VOD has a spatial resolution of 0.1° (~10 km) and nearly daily temporal resolution (Parinussa et al., 2014). The time period covered by AMSR-E is from 2 June 2002 to 3 October 2011, but we used a 9 year range from 4 July 2002 to 26 June 2011 in our analysis by excluding the beginning and the end of the AMSR-E records. The same time period is also used for the MOD09A1 NDVI dataset.

VOD data, which has a near daily temporal resolution, is converted to an 8 day average product to reduce noise and ensure complete coverage over Australia (10° S to 45° S and 110° E to 160° E) per temporal interval. The grid cells with radio frequency interference (RFI) are excluded from our analysis. RFI is caused by man-made transmitters, such as radars and wireless communications. These transmitters can be operated in the same frequency range as passive microwave observation, including VOD. Thus, natural signals captured by passive microwave observations are sometimes contaminated with RFI (Nijs et al., 2015).

An example comparison time series of VOD and NDVI from July 2002 to June 2011 at one of the observed DOC sites, Silent Grove, WA (17.131° S, 125.374° E) can be seen in Fig. 1. It is shown that both VOD and NDVI have a similar seasonal cycle. Vegetation types that are present within the VOD (0.1°) and NDVI (0.005°) pixel influence the differences in VOD and NDVI behaviour. The difference between VOD and NDVI spatial resolution can be clearly seen in the example 2° by 2° spatial maps around the Silent Grove area (Fig. 1). We use both VOD and NDVI together to combine their strengths.

The global 0.05° land cover map based on the MODIS MCD12C1 product is used for classifying the dominant land cover type within each VOD pixel. The land cover classification system is as proposed by the University of Maryland (UMD scheme) (Hansen et al., 2000). Figure 2 shows the 0.05° land cover type map of Australia, with observed curing sites marked with crosses. The land cover map used here is from year 2010.

A burned area product from MODIS is acquired for further evaluation of the recalculated GFDI from satellite based DOC results. The monthly archived MODIS burned area map reprojected for Australia is obtained from Remote Sensing at the NCI site (Paget and King, 2008). There are two separate MODIS burned area products: the MCD45A1 and the MCD64A1. The MCD64A1 burned area product is preferred over MCD45A1, since it was proven to be more accurate (Andela and van der Werf, 2014; Padilla et al., 2015; Ruiz et al., 2014). Its spatial specification is exactly the same as the MODIS reflectance

dataset, with temporal availability from August 2000 onwards. To ensure high quality of the burned pixels, only pixels with the valid data flag from the provided quality control file are included in the analysis. Over 99 % of pixels from mid-2002 to mid-2011 are classified as unburned. To reduce the number of prescribed burned and other low power anomalies detected by the burned area product, a fire radiative power (FRP, unit: MW) from MODIS active fire product (MCD14ML) is used to mask out low severity fires.

**2.2 Ground Based Curing Observations**

The observed grassland DOC data was provided by Bushfire and Natural Hazards Cooperative Research Centre and its partner agencies [Project reference: http://www.bushfirecrc.com/projects/a14/grassland-curing]. The observed data were collected from several sites across Australia and New Zealand, ranging from August 2005 to March 2009, usually during summer (Fig. 2). Selected sites were intended to represent broad coverage of major grassland types. Note that the number of locations and

samples taken were highly dependent on the availability of field observers from fire management agencies; data were collected with inconsistent interval between data collection dates (Anderson et al., 2011). Three types of data collection approaches were used: visual estimation, levy rod method, and destructive sampling. Due to the number and availability of data as well as their accuracy, only observed DOC from the levy rod method was used in this study. Anderson et al. (2011) and Newnham et al. (2011) state that the levy rod measurement is reliable with RMSE of 13.5 % and a bias of less than 1 % when compared with

destructive sampling.

To identify robust relationships between the site observed and remotely sensed DOC, a number of site criteria must be met. Sites meeting these criteria were used for calibration of the VOD and NDVI satellite based DOC models, while all of the valid records were used for evaluation. One major factor in deciding the site selection is the land cover properties of the observed DOC site. The 0.05° land cover type map (MCD12C1) is used for classifying the site location land cover (Hansen et al. 2000).

Since 0.1° VOD pixel is covered by 2 by 2 0.05° land cover pixels, the corresponding 2 by 2 pixels of land cover type for each observed curing site can be acquired. The land cover type and homogeneity of each observed DOC site can then be determined, where the site is considered to have a homogeneous land cover only if all four land cover pixels corresponding to the VOD pixel are the same. In case of a site with heterogeneous land cover type, the dominant land cover with the most pixels out of four will be considered as the representative land cover. All observed DOC sites can be categorised into the following land

cover types: evergreen broadleaf forest, open shrubland, savannas, woody savannas, grasslands, croplands, and urban.

According to the land cover information for each 0.1° VOD pixel, sites identified as evergreen broadleaf forest pixels were removed from the analysis. There are three out of 37 sites situated in the evergreen broadleaf forest, which are Darnum, VIC (mixed grass), Simcocks, WA (improved pasture), and Neerim South, VIC (mixed grass). Even though actual locations of all observed curing sites were in grassy areas, the VOD signal is a mixture of grassland and forest when the sites are surrounded

by dense forests within the same 0.1° pixel.

All sites were also examined to ensure a negative correlation between VOD and the in situ DOC data. That is, since VOD is a proxy for water content in above ground biomass, an overall negative correlation between VOD and curing is expected. If this

is not the case, then there is likely some other activity within the 0.1° pixel that disrupts this basic relationship; this effect was found in three sites (Durran Durra, NSW (native grass), Monaro, ACT (improved pasture), and Parry Lagoons, WA (native grass)). The sites without the negative correlation with VOD also had no correlation with NDVI suggesting other land cover types where dominating the signal. Thus, six out of 37 sites are excluded from the analysis.

In addition, there are eight sites (Umbigong, ACT (native grass), Kilcunda, VIC (improved pasture), Tooradin, VIC (improved pasture), Tooradin North, VIC (improved pasture), Caldermeade Park, VIC (improved pasture), Kaduna Park, VIC (improved pasture), Hobart Airport, TAS (native grass), and Jerona, QLD (native grass)) in which VOD data are not available. Most of these are due to sites being located near the coast or a large body of water, where the VOD signal is strongly influenced by the water itself. With the remaining 23 out of 37 sites, several site selection criteria were applied for the calibration phase. The criterion used here to maintain consistency in observation time series requires sites to have at least eight consecutive records, where records are considered consecutive when they are separated by no more than 15 days. Only the consecutive series of records within the selected sites are included in the analysis for the calibration phase. This ensures that the derived model contains the temporal evolution of DOC within years. Only five out of 23 sites are retained for this group, containing a total of 122 (out of 238 total) observations. The selected sites are: Majura, ACT (improved pasture), Tidbinbilla, ACT (mixed grass), Ballan, VIC (improved pasture), Murrayville 1, VIC (native grass), and Murrayville 2, VIC (improved pasture). Multiple linear regression models of VOD and NDVI were then calibrated with the observed curing from the final selected sites.

**2.3 Meteorological Datasets**

To further assess the usability of the satellite based curing acquired from the VOD and NDVI model, the GFDI is computed. Additional meteorological data needed for GFDI computation are dry bulb or maximum temperature, 3 pm relative humidity, maximum wind speed, and fuel load (Purton, 1982). Since fuel load is often set as a constant value of 0.45 kg m$^{-2}$ (Sharples et al., 2009a), there are 3 remaining input datasets needed. These gridded, meteorological datasets are usually derived from the network of ground observation stations across Australia. The range for these datasets is from 4 July 2002 to 26 June 2011 to exactly match with the VOD 9 year range. Both temperature and relative humidity datasets are acquired from the Australian Water Availability Project (AWAP) (Jones et al., 2009). Note that relative humidity is derived from vapour pressure and temperature data. These AWAP datasets have a 0.05° spatial and daily temporal resolution with a coverage region of 10° S to 44.5° S and 112° E to 156° E. For maximum wind speed data, the reanalysis maximum daily wind speed is computed from the ERA–Interim wind components dataset, acquired from the European Centre for Medium–Range Weather Forecasts (ECMWF) (Dee et al., 2011). The reanalysis wind components dataset is available globally at approximately 0.8° spatial resolution at a 6 hour interval.

**3. Methods**

**3.1 Developing VOD NDVI based dynamic DOC estimates**

As indicated by past studies (Dilley et al., 2004; Peterson et al., 2008), NDVI has a significant relationship with live fuel moisture content and DOC. In addition, the NDVI dataset has a spatial resolution of 0.005°. Thus, we investigate whether VOD adds any information to satellite based DOC beyond that embodied in the NDVI through the use of a multiple linear regression model. In addition, a past study suggested that a modified form of NDVI, referred to as relative greenness (RG), has stronger relationship with DOC than NDVI (Newnham et al., 2011). One of the two alternatives Newnham et al. (2011) proposed is the range based RG, which can be computed by the following Eq. (2):

$$RG = \frac{NDVI - NDVI_{min}}{NDVI_{max} - NDVI_{min}} \qquad (2)$$

where the $NDVI_{min}$ and $NDVI_{max}$ are the minimum and maximum NDVI value over a specified time range. Note that we did not attempt to compare the other purposed alternative, the spread based RG, but only range based (per pixel) RG. In Newnham et al. (2011), while range based RG performance is not as good as preferred spread based RG ($r^2 = 0.62$ and RMSE = 14.2 %), it is still better than plain NDVI (NDVI had $r^2 = 0.50$ and RMSE = 16.4 %, while 2.5 years range based has $r^2 = 0.57$ and

RMSE = 15.1 %). Note that while we cannot exactly reproduce 10 years time range Newnham et al. (2011) used, since our study time frame is 9 years, we tried various 2.5 years time ranges that overlapped with Newnham et al. (2011) study period. However, given the same observed DOC data and NDVI dataset, we were unable to reproduce a result where the RG correlation with DOC is stronger than NDVI. Further analysis showed that the RG results were very sensitive to the selected time range for the computation, such that results were inconsistent with relatively small differences in the selected range. Due to this, RG

is not used in this study and NDVI is used directly in forming a multiple linear regression model to estimate satellite based curing. Some experimentation revealed that the VOD anomalies, computed from the difference between VOD and average VOD over a specified temporal range, yields the best correlation with DOC, but only if the VOD anomalies are computed from the range matching the in situ DOC observation range for each specific site. The range selection for computing VOD anomalies can be quite problematic, since it can heavily influence the correlation result, and no pattern could be found for

determining an appropriate VOD range for any other locations outside the observed DOC sites. Thus, we focus our analysis on using the absolute VOD value. The linear regression equation for curing and VOD correlation can be expressed as Eq. (3):

$$C = x_1 + x_2(VOD) \tag{3}$$

where $x_1$ and $x_2$ are the intercept and slope of the relationship.

Utilising both VOD and NDVI datasets, the following multiple linear regression equation for estimating DOC can be expanded

from Eq. (3) as Eq. (4):

$$C = x_1 + x_2(VOD) + x_3(NDVI) + x_4(VOD)(NDVI) \tag{4}$$

where $x_1$ to $x_4$ are the intercept and coefficients of VOD, NDVI and the product of VOD and NDVI (interaction term), respectively. Using a stepwise regression, the calibrated final model with corresponding coefficients can be determined. The stepwise fit algorithm used here selects the significant terms with the lowest p-value, which is smaller than the entrance

tolerance, to be included in the model first. Next, the algorithm chooses the next most significant term that is still less than the entrance tolerance. This process is repeated until either there are no remaining significant terms or all terms are included in the final model (Draper and Smith, 1998). After the final model is calibrated, we evaluate the DOC model with all valid (23) observed DOC sites.

## 3.2 Comparing with existing DOC estimates

We acquire existing satellite based DOC products available from Bureau of Meteorology and compare their performance with our model. There are five models available, four are based on Newnham et al. (2010) and one is based on Martin et al. (2015) studies. We decided to test only one of Newnham's models – the one with the best overall RMSE (Method B), and Martin's model (MapVic). Both Method B and MapVic DOC models are as described in Eq. (5) and Eq. (6), as shown below:

$$C_{Method\ B} = 237.31 - 190.14(NDVI) - 142.66(\frac{\rho_7}{\rho_6}) \tag{5}$$

$$C_{MapVic} = 113.80 - 88.41(NDVI) - 67.71(GVMI) \tag{6}$$

where $\rho_6$ and $\rho_7$ are spectral reflectance band six and seven from MODIS reflectance dataset and GVMI is Global Vegetation Monitoring Index (Martin et al., 2015; Newnham et al., 2010). GVMI can be calculated by:

$$GVMI = \frac{(\rho_2 + 0.1) - (\rho_6 + 0.02)}{(\rho_2 + 0.1) + (\rho_6 + 0.02)} \tag{7}$$

where $\rho_2$ and $\rho_6$ are spectral reflectance band two and six from MODIS reflectance dataset (Ceccato et al., 2002).

To compare both Method B and MapVic model performance with our model, we evaluate them using the same observed DOC sites and evaluation methods. We also computed recalculated GFDI with both Method B and MapVic DOC and assess their burned area prediction capability.

## 3.3 Comparing GFDI dynamics using different DOC estimates

Several revisions of GFDI were made by past studies (Noble et al., 1980; Purton, 1982). The GFDI revision used in this paper is modified Mark 4 GFDI, since it is the grassland fire danger assessing system that is generally being used by Bureau of Meteorology (Sharples et al., 2009b). Originally, the fire danger rating system was presented in a circular slide rule. A mathematical equation representation of modified Mark 4 GFDI was derived from the circular meter, and can be expressed as follows Eq. (8) (Purton, 1982):

$$\text{GFDI} = \text{Q}^{1.027}\text{f(C)}\exp(-1.523 + 0.0276\text{T}_{max} - 0.2205\sqrt{\text{H}_{3pm}} + 0.6422\sqrt{\text{V}_{max}}) \tag{8}$$

where Q is the fuel load (kg m$^{-2}$), $\text{T}_{max}$ is the dry bulb or daily maximum temperature ($^{\circ}$C), $\text{H}_{3pm}$ is the daily relative humidity at 3 pm (%), $\text{V}_{max}$ is the daily maximum wind speed (km h$^{-1}$), and f(C) is the curing factor. The curing factor can be calculated by Eq. (9):

$$\text{f(C)} = \exp(-0.009432(100 - \text{C})^{1.536}) \tag{9}$$

where C is the grassland DOC (%).

The GFDI is computed on the basis of meteorological input data and either a constant DOC at 100 % or satellite based dynamic curing values. These different GFDI datasets along with the burned area data (MCD64A1) can be used to examine the changes due to variable DOC spatially and temporally. By pairing up burned and unburned pixels with their associated GFDI pixel, we can assess the number of burned and unburned pixels for each GFDI severity level. Using histogram and receiver operating characteristic (ROC) analysis, the difference between original GFDI with constant DOC at 100 % and recalculated GFDI with satellite based dynamic DOC can be assessed (DeLong et al. 1988; Zweig and Campbell 1993).

## 4. Results

### 4.1 Comparing VOD-NDVI based and existing DOC estimates

Across all selected observed DOC sites (excluding the forest areas) from July 2002 to June 2011, the $r^2$ of VOD and NDVI is 0.52 with an RMSE of 0.11. Using the linear model, as described by Eq. (3), the DOC and VOD correlation result has a significant relationship and an $r^2$ of 0.20 with RMSE of 20.80 %. The scatter plot showing the correlation between VOD, NDVI and combined VOD and NDVI terms are as shown in Fig. 3, while Fig. 4 shows the residual DOC unexplained by NDVI (differences between observed DOC and NDVI-based DOC) against VOD and combined VOD and NDVI terms.

However, at this level of $r^2$, VOD alone is not reliable enough to estimate DOC, especially across Australia in general. The study of Newnham et al. (2011) indicated that NDVI alone can perform better at estimating DOC, with an $r^2$ of approximately 0.50 for a DOC and NDVI linear relationship. The combined explanatory power of NDVI and VOD is explored using a multiple linear regression analysis, as expressed by Eq. (4). The first final model includes the VOD and NDVI interaction ($x_4$) and NDVI ($x_3$) terms, and is as shown in Eq. (10). The calibrated $r^2$ for this model is 0.67 with RMSE of 13.40 %. VOD was excluded as a predictor in the first final model, as expressed in Eq. (10), because during the stepwise regression, when the NDVI and (VOD)(NDVI) terms are included as the first and second predictors, the VOD term does not contribute in improving the final model prediction (i.e. p-value exceeds the acceptance threshold, preventing overfitting). When the NDVI term is excluded, the (VOD)(NDVI) term is included first, followed by the VOD term, as expressed in the second final model, shown in Eq. (11). The second final model has a calibrated $r^2$ of 0.54 and RMSE of 15.95 %. Table 1 shows the correlation results for both models.

$$\text{C}_1 = 145.57 - 260.82(\text{NDVI}) + 137.19(\text{VOD})(\text{NDVI}) \tag{10}$$

$$C_2 = 48.70 - 147.60(VOD) + 259.95(VOD)(NDVI) \tag{11}$$

The DOC models are then evaluated with all (23) valid observed DOC sites and independent (18) observed DOC sites (excludes 5 sites that were used in calibration). The evaluation results for the first model are also shown in Table 1, where the evaluated $r^2$ is 0.55 and 0.44 with RMSE of 15.25 % and 16.76 % for all sites evaluation and independent sites evaluation, respectively. The second model evaluations results have $r^2$ of 0.50 and 0.54, with RMSE of 15.95 % and 15.53 % for all sites evaluation and independent sites evaluation, respectively. While the evaluations resulted in degradation in model performance over the calibration in most cases, the independent evaluation of the second model has a slightly better evaluation performance. These results can be compared to those obtained using existing remotely sensed DOC estimates which are also shown in Table 1. The MapVic DOC has a lower $r^2$ and higher RMSE, while the Method B DOC has a higher $r^2$ and lower RMSE when compared with both of our models in all sites evaluation. This indicates that Method B has the best evaluation among the three models, while MapVic is the worst, and our models sit in the middle between the two. However, during an independent sites evaluation, both models with VOD have the worst performance (lowest $r^2$ and RMSE). This result is not entirely surprising as all the observations used here were also used in the calibration of Method B (Newnham et al., 2010), a subset is used in the calibration of our method, and MapVic was developed using an independent visual estimates dataset. That is, there is no independent data available for testing Method B, while both our method and MapVic are being tested against independent data. While our first and second models do not have obvious advantages over one another, since the second model only performs better than the first in independent sites evaluation, we decided to pick the first model as our representative model for further comparison with the existing models from this point, since the terms in the first model were selected based on stepwise fit regression with none of our interference (we intentionally removed NDVI term in the second model before applying the stepwise fit regression).

Using the relationship between VOD, NDVI, and observed DOC from the first model, as stated in Eq. (10), we calculated satellite based DOC for Australia. Figure 5 presents maps of satellite based DOC data averaged over the summer periods (December, January, February, DJF) for the years 2002–2003 and 2010–2011. From mid-2002 to mid-2011, the overall average curing for the Australian summer period is the highest during 2003 and the lowest during 2011. Note that the pixels that are classified as any forest types are masked out in white. Comparison time series between satellite based and site observed DOC at Silent Grove, WA (same location as shown in VOD and NDVI example comparison in Fig. 1) is also shown at the top of Fig. 4 as an example.

To determine the amount of spatial variation in DOC across Australia, we computed the standard deviation of all valid DOC estimates across the continent within a single time step. All areas that are indicated as forests by the land cover type map are excluded from the analysis. The spatial variation time series can then be plotted for the available time period of mid-2002 to mid-2011, as shown in Fig. 6. Note that the continental mean spatial DOC standard deviation is 20.39 %. This indicates that there is significant spatial variability in DOC that persists across all years, and contains a small seasonal component. For a normally distributed variable, 95 % of values would lie within two standard deviations, which is ±40.78 % in this case. Further analysis on DOC spatial standard deviation are as shown in Table 2. This includes seasonal, monthly, and land cover type spatial standard deviation of DOC. From both seasonal and monthly spatial standard deviation of DOC, it is shown that DOC has the highest spatial variation during winter, which is especially true for northern Australia (Anderson et al., 2011).

In addition, based on time series of satellite based curing data, Fig. 7 reveals the spatial distribution of standard deviations calculated for each pixel. It shows that most of the strong temporal variation occurs in the south, especially in the southeast and southwest of Australia. Several areas in the midcontinent that have unexpectedly high variation are likely due to rare inundation events. The continental mean temporal standard variation is at 11.88 %. Together, Fig. 6 and Fig. 7 show the variability in DOC that will impact calculations of fire danger indices.

## 4.2 Comparing GFDI dynamics using different DOC estimates

The spatial plot for maximum summer recalculated GFDI from the DOC multiple linear regression model is shown in Fig. 8, where the top row (a) and (b) are the maps for summer 2003 and summer 2010, respectively. The magnified regions for example fire events in Weston Creek, ACT 2003 and Toodyay, WA 2010 events can be seen in the bottom row (c) and (d). The fire locations are marked with a red crosshair. White pixels are forest areas that were masked using the land cover map. Overall, summer 2003 has 4.51 % more areas indicated as severe or higher GFDI than summer 2010. MCD64A1 burned area map (Fig. 9), also suggested that summer 2003 had 91.45 % more severe wildfire counts than summer 2010. It should be noted that high GFDI values do not guarantee a fire as there is no accounting for ignition sources, rather a higher GFDI value indicates higher probability of fire ignition and that if a grassland fire were to start it would spread faster compared to low GFDI values, given no fire suppression activity. Further complicating comparison of Fig. 8 and Fig. 9 is the presence of prescribed burns that are deliberately done during low to moderate GFDI conditions, and that some fires shown in Fig. 9 occur in forested areas where GFDI is not applicable. Nevertheless, they provide a picture of the inter-annual spatial variability in both GFDI and burned area.

The time series plots of recalculated GFDI at Weston Creek, ACT, and Toodyay, WA, for the example 2003 and 2010 fire events were produced and are shown in Fig. 10. The black line represents the recalculated GFDI from variable DOC, while the dashed, light green line is for original GFDI with constant DOC at 100 %. These locations are marked with red crosshair indicators on the spatial maps (Fig. 8). Note that the original GFDI time series peaks every year, whereas the recalculated GFDI with variable curing time series shows sudden peaks in the days near major fires. The Weston Creek fire was part of the 2003 Canberra bushfire complex, where multiple fires merged and rapidly propagated from 18–22 January 2003, burning 1,600 km$^2$ (McLeod, 2003). The weather conditions on 18 January 2003 were extreme with temperature as high as 40° C and wind exceeding 60 km h$^{-1}$. The Toodyay fire was much smaller in magnitude, burning just over 30 km$^2$ on 29 December 2009. The Weston Creek area is mostly comprised of forest with mixed land cover, whereas the Toodyay area is mostly a mix between croplands and savannas.

Using a burned area observation dataset from MODIS (MCD64A1), we test the effectiveness of GFDI with variable curing in increasing the probability that fires will occur in high GFDI severity levels compared to the probability that fires will occur in low–moderate GFDI severity levels. Low intensity fires, such as prescribed burned, are removed from the burned area observation by using the FRP provided in MODIS active fire product (MCD14ML) to mask out burned area that have low FRP. We assume any burned area with FRP lower than 100 MW to be unburned (associated with low–moderate GFDI risk). At each burned and unburned daily data point, the corresponding daily GFDI was calculated. The GFDI histogram in Fig. 11 shows the frequency of satellite based recalculated GFDIs and constant based (DOC = 100 %) reference GFDIs over burned and unburned areas. Figure 11 shows that the recalculated GFDI places the largest percentage of unburned pixels in the low–moderate GFDI severity class, with ~75 % of all unburned pixels occurring in the low–moderate or high severity classes. Meanwhile the reference (DOC = 100 %) GFDI places ~80 % of unburned pixels in the high, very high, severe and extreme classes.

We can evaluate the performance in correctly assigning burned and unburned area for both recalculated and reference GFDI by using the concept of ROC, as described earlier. Assume that the MCD64A1 burned area map represents the true condition and that the GFDI severity level represents the predicted condition, where the prediction is positive when GFDI level is classified as high or above for a burned area and low–moderate for an unburned area. Table 3 shows the contingency table, including both type I (unburned area with high or above GFDI level; false positive) and type II (burned area with low–moderate GFDI level) errors. Though recalculated GFDI has a lower true positive rate of correctly assigning burned area than reference GFDI (0.86 vs 0.95), it is much better at assigning unburned area correctly, i.e. lower false positive rate (0.38 vs 0.53). Overall accuracy for recalculated GFDI is higher than the reference GFDI (0.62 vs 0.47).

Both Method B and MapVic DOC are then used to compute recalculated GFDI and compare with burned area observation dataset in the same manner as our DOC model. From the ROC analysis in Table 4 for Method B and MapVic recalculated GFDI, we found that even though Method B has the best DOC evaluation results (highest r$^2$, lowest RMSE) and highest overall recalculated GFDI burned and unburned detection accuracy at 0.84, it is the worst at detecting burned area correctly with a true positive rate of only 0.10. This concurs with the findings in Newnham et al. (2010) who found Method B to consistently under predict DOC, and hence it produces fewer cases of high or above GFDI severity. Our model is in the middle ground between Method B and MapVic in terms of overall accuracy.

## 5. Discussion

### 5.1 Evaluation of Satellite based DOC

Previous studies that derived satellite based DOC have mostly relied solely on NDVI as a predictor. In the study by Newnham et al. (2011) various forms of NDVI were used, including a straight NDVI linear regression and relative NDVI, as shown in Eq. (2). Their results suggested that a linear regression model based on NDVI alone reproduced site observations with an r$^2$ of 0.50. The results presented here indicate that inclusion of VOD in the regression model yields a comparable performance with r$^2$ of 0.44 to 0.55 (with independent sites evaluation at the worst and all sites evaluation at the best performance). However, when we compare the evaluation results with the currently available satellite based DOC from Bureau of Meteorology, the best performer is Method B in both r$^2$ and RMSE for both evaluation methods (Newnham et al., 2010). We note that this is not a fair comparison as all the data used to evaluate the models was used to calibrate Method B, regardless of our evaluation methods. MapVic (Martin et al. 2015), on the other hand, was developed using a totally independent visual assessment dataset and hence is being evaluated against an independent dataset, regardless of our evaluation methods. Our first model performs better in all sites evaluation (include subset of the data for calibration) than independent sites evaluation, as expected. However, our second model performs better in independent sites evaluation than all sites evaluation. Regardless, the overall performance is still not obviously better than neither Method B or MapVic models. While adding VOD to the DOC estimation model may introduce a comparable alternative to the existing optical based models, there is still no clear advantage of including VOD over the current methods.

An earlier study for estimating DOC directly with NDVI yielded even smaller RMSE of up to 6.3 %, but that particular study is focused on data from only three different sites, within a limited study area of 1 km$^2$ (Dilley et al., 2004). Older studies that have used NDVI data derived from the National Oceanic and Atmospheric Administration's (NOAA) Advanced Very High Resolution Radiometer (AVHRR) have the same problem of interference from clouds and atmospheric effects, but do not have the advantage that VOD is not affected by clouds or aerosol interference.

### 5.2 GFDI with Various DOC estimates

Though the overall recalculated GFDI from Method B DOC is the best (overall accuracy from best to worst is 0.85 for Method B, 0.67 for our DOC model, 0.61 for MapVic, and 0.48 for GFDI with 100 % constant DOC), we found that it is the worst at detecting burned area correctly (true positive rate from best to worst is 0.88 for GFDI with 100 % constant DOC, 0.74 for MapVic, 0.69 for our DOC model, and 0.09 for Method B). Our VOD and NDVI based DOC model (first model) has a good balance in having the second best evaluation result and overall recalculated GFDI accuracy with a decent correct burned area detection rate. We note that GFDI is only an indicator for the level of fire risk, and does not guarantee that a fire will occur, even at an extreme danger level. However, the improvement in accuracy indicates that inclusion of time and space varying DOC estimates makes it much more likely that areas identified at a GFDI severity level of high or above will burn than a low–moderate severity level area.

**5.3 Limitations**

It is worth noting here that in an operational setting atmospheric interference by clouds or smoke will cause gaps in the optical (NDVI) data, though the VOD data remains unaffected. We also note that while the VOD data used here was derived from the AMSR-E sensor, which is no longer operational, VOD data derived from currently operating passive microwave sensors, such as the Advance Microwave Scanning Radiometer 2 (AMSR2), could be used in an operational setting. It should also be noted that VOD's moderately coarse resolution of 0.1° may not be fine enough for use in many applications.

Reducing the chance of incorrectly assigning unburned and burned areas correctly from the ROC analysis made here is purely based on using the burned area map as a true baseline. However, the burned area map may include fires that are deliberately lit in low–moderate conditions, such as prescribed burns and fires that the GFDI is not designed for, such as a fire that burns in forested regions. Prescribed burns and low intensity fires are however minimised by applying low FRP threshold, using information from the MODIS active fire product. The ROC analysis result here is only used to reinforce the idea that using the reference GFDI with constant curing (100 %) leads to overestimating GFDI in some situations, and might result in misleading fire danger warnings.

The satellite based DOC produced here is also at a moderate spatial resolution, which is a limitation of many satellite products. However, DOC in reality can vary over spatial scales much finer than the satellite footprint (less than 500 m). As such, our model should only be used as a guide for dynamic, near daily assessment of grassland curing at coarse to moderate spatial scales. This is also true for other satellite based DOC models, including Method B and MapVic models.

**6. Conclusions**

This study developed an alternative approach for estimating the grassland DOC using a relationship between the observed DOC and satellite based VOD and NDVI. The satellite based dataset was evaluated against the observed DOC data, which resulted in a comparable performance with the currently existing optical based DOC estimation models. Despite the relatively coarse spatial resolution and temporal coverage of VOD and NDVI datasets used in this study, the satellite based DOC dataset produced from our model has the potential to contribute to the preparedness of fire management agencies and improve fire spread modelling. With a comparable, and arguably more balanced, performance in correctly predicting burned and unburned area through GFDI than currently available satellite based DOC models (i.e. Method B and MapVic), our model could provide an appealing alternative estimated DOC data for GFDI computations and fire risk modelling.

**Data availability**

The following datasets and their associated sources or contact points are as listed below:

- MODIS MOD09A1, MCD16C1, MCD14ML, and MCD64A1 for NDVI computation, land cover type map, active fire product, and burned area map for Australia are freely available from NASA via Remote Sensing at NCI (mosaicing and regridding by CSIRO): http://remote-sensing.nci.org.au/u39/public/html/modis/lpdaac-mosaics-cmar/
- AMSR-E VOD dataset for Australia is available upon request by contacting Yi Liu: yi.liu@nuist.edu.cn
- Method B and MapVic DOC products information can be found at: http://data.auscover.org.au/xwiki/bin/view/Product+pages/Grassland+Curing+MODIS+BoM and the datasets are downloaded from the following catalogue: http://opendap.bom.gov.au:8080/thredds/catalog/curing_modis_500m_8-day/aust/netcdf/catalog.html

- Observed DOC dataset via visual assessment, levy rod, and destructive methods is available upon request from the following Bushfire and Natural Hazards Cooperative Research Centre legacy project: http://www.bushfirecrc.com/projects/a14/grassland-curing
- Maximum daily observed gridded temperature and vapour pressure dataset for Australia are freely available from AWAP: http://www.bom.gov.au/jsp/awap/
- ERA-Interim maximum daily reanalysis gridded wind components for Australia are freely available from ECMWF: http://www.ecmwf.int/en/research/climate-reanalysis/era-interim

## Author contribution

Wasin Chaivaranont, Jason Evans, and Yi Liu are involved in initial research planning and experiments design. For the rest of the study, including the analysis of the results, paper writing, and manuscript proofreading, all authors (Wasin Chaivaranont, Jason Evans, Yi Liu, and Jason Sharples) are involved.

## Competing interests

The authors declare that they have no conflict of interest.

## Acknowledgements

We would like to acknowledge the following parties:
- NASA for making MODIS data freely available.
- NCI and CSIRO for making a freely available preprocessed MODIS data for Australia.
- Bureau of Meteorology for making freely available satellite based DOC dataset and continue to generate and distribute the data products set up by the AWAP project.
- Bushfire and Natural Hazards Cooperative Research Centre and its partner agencies for allowing us to use their observed DOC dataset from their legacy project.
- AWAP for making daily Australia climate gridded dataset freely available.
- ECMWF for making ERA-Interim reanalysis gridded dataset freely available.

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

**Figures and Tables**

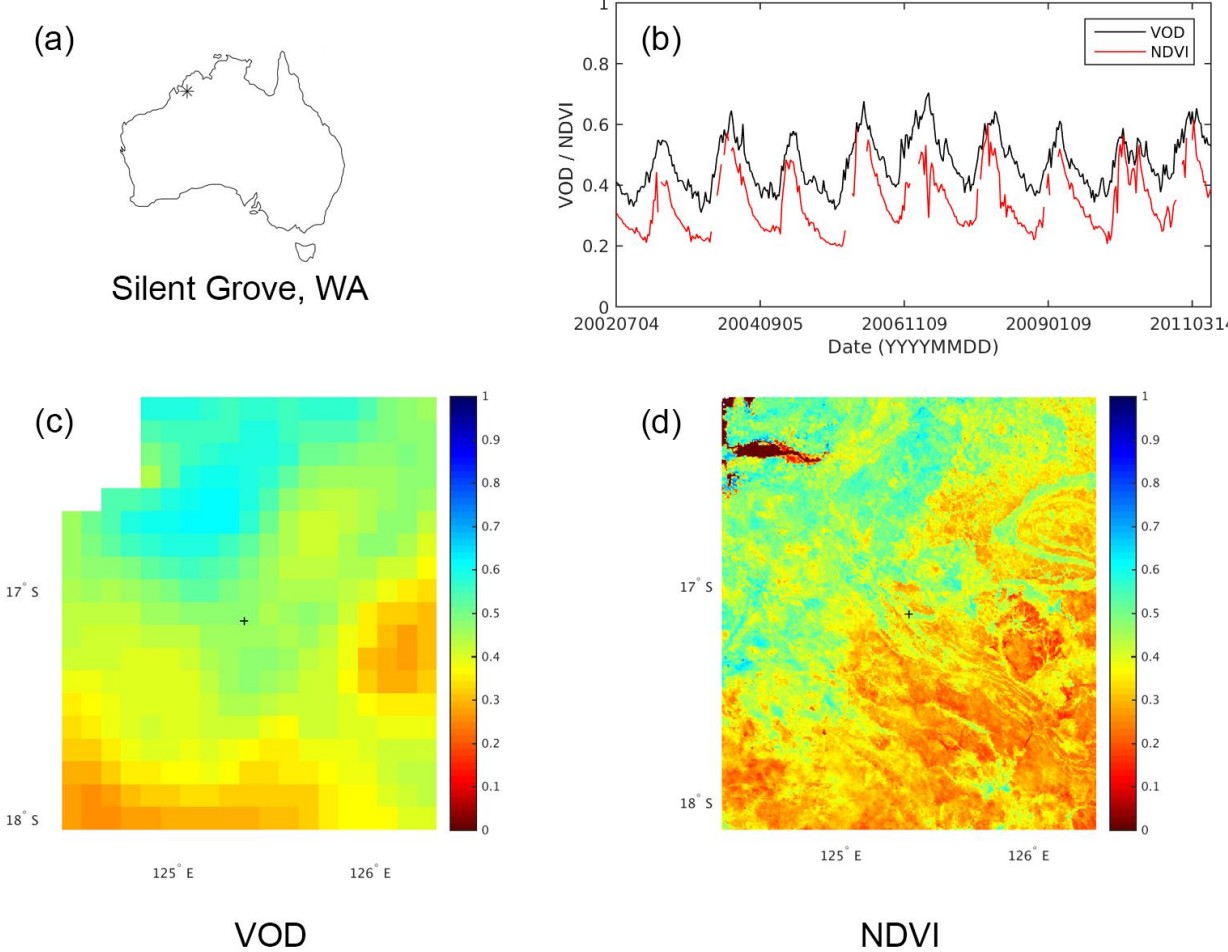

**Figure 1: Example Vegetation Optical Depth (VOD) and Normalised Difference Vegetation Index (NDVI) time series (b) and spatial maps, (c) for VOD and (d) for NDVI, at Silent Grove, WA (17.131° S, 125.374° E). The star (*) indicate the location of the time series on Australia map (a).**

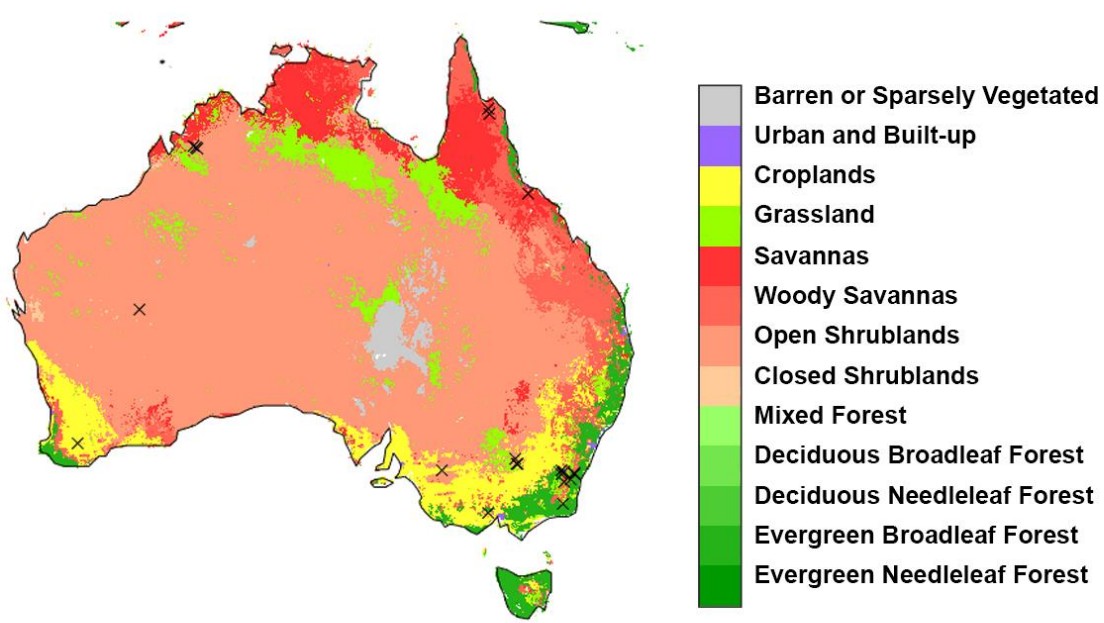

**Figure 2: MCD12C1 land cover type map for Australia (Hansen et al., 2000). The locations of 23 valid observed degree of curing (DOC) sites are marked with crosses.**

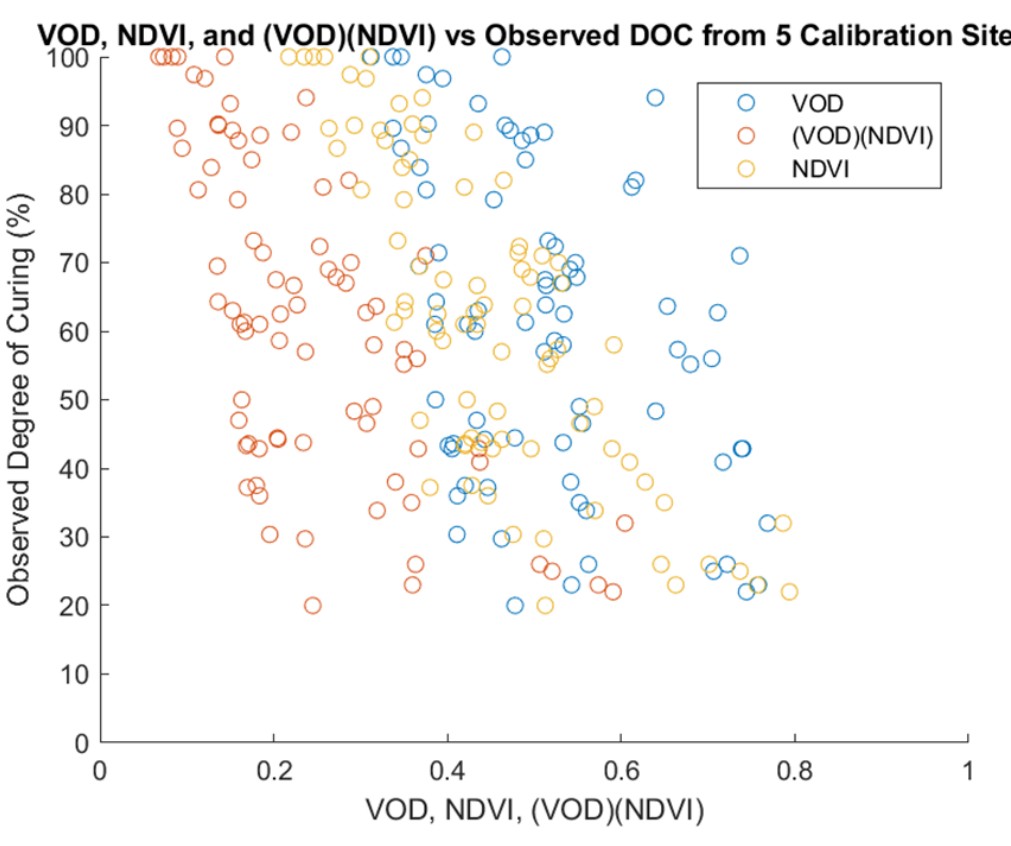

5      **Figure 3: Scatter plot of observed degree of curing (DOC) from five calibration sites against VOD, NDVI, and combined VOD and NDVI terms.**

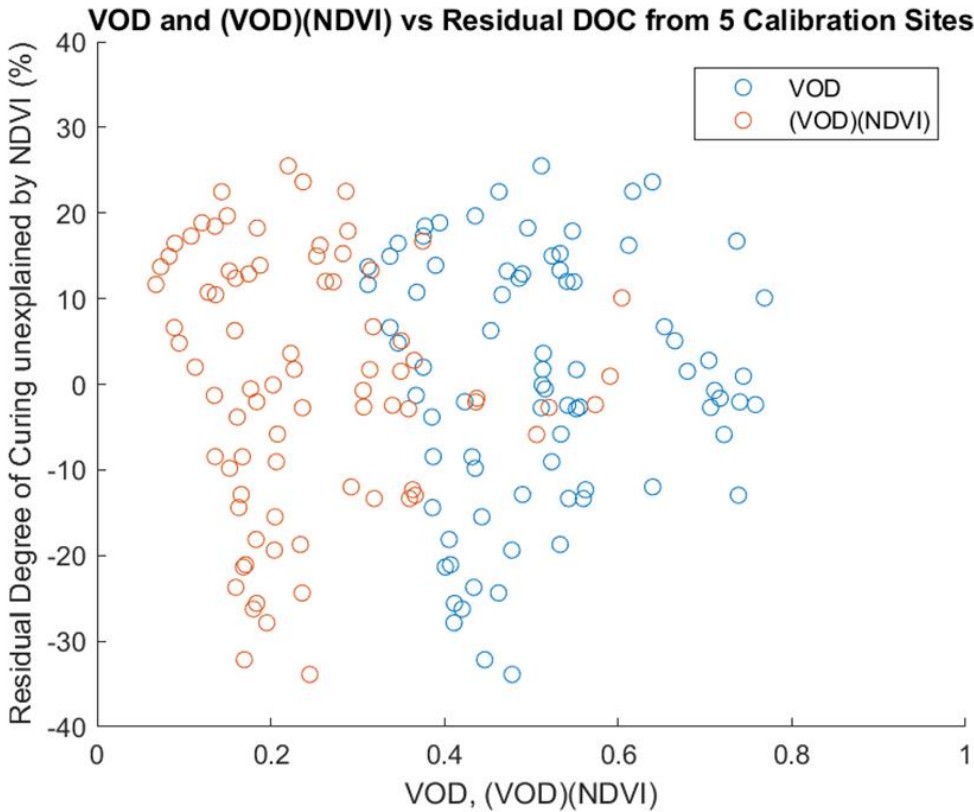

**Figure 4: Scatter plot of residual observed degree of curing (DOC) from five calibration sites that are unexplained by NDVI against VOD and combined VOD and NDVI terms.**

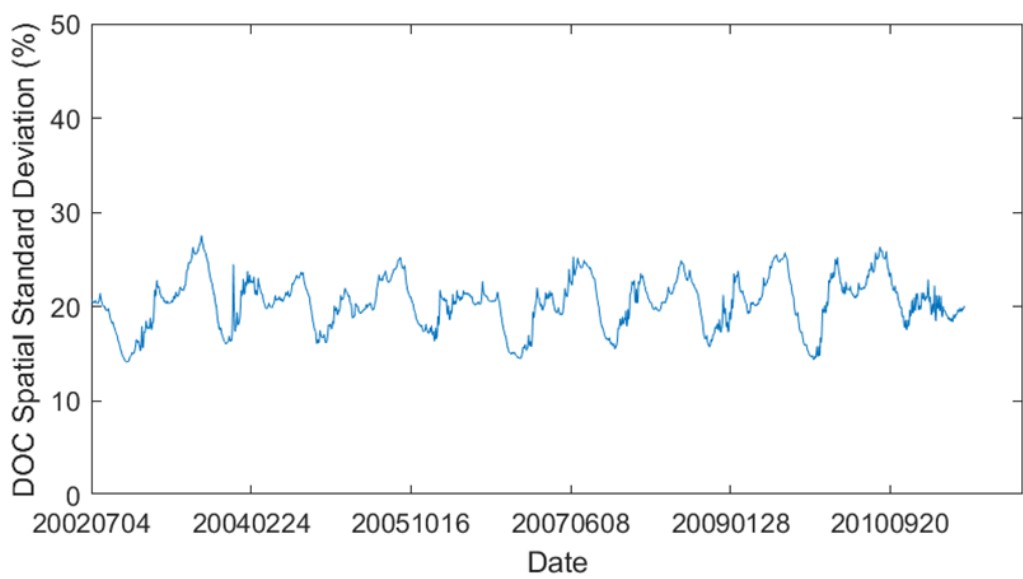

**Figure 5: Example satellite based and site observed degree of curing (DOC) time series comparison at Silent Grove, WA (17.131° S, 125.374° E) (b), where the star (\*) indicate the location of the time series on Australia map (a). Satellite based DOC across Australia during summer (December, January, February) for 2002—2003 (c) and 2010—2011 (d) are shown with forest areas masked out.**

**Figure 6: Spatial standard deviation of estimated degree of curing (DOC) time series from 4 July 2002 to 26 June 2011.**

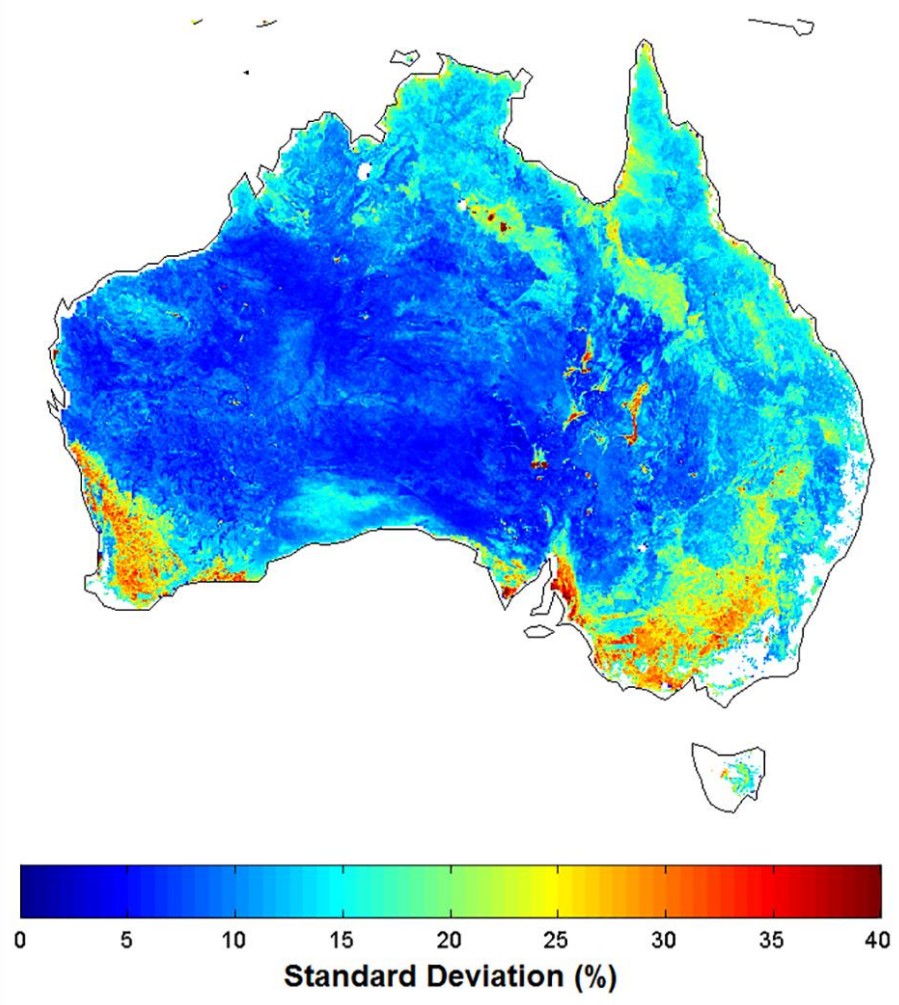

**Figure 7: Temporal standard deviation of estimated degree of curing (DOC) map from 4 July 2002 to 26 June 2011.**

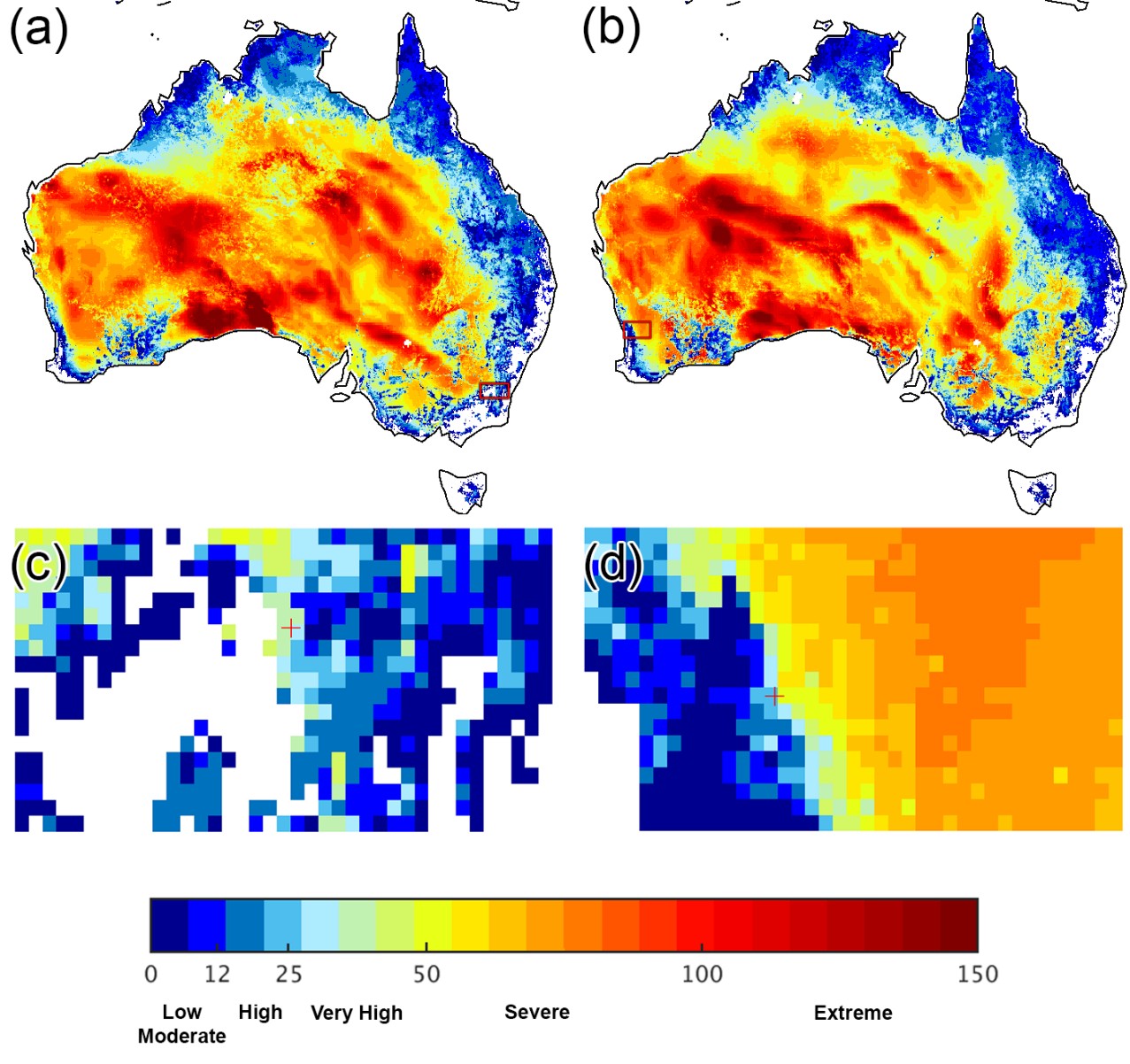

**Recalculated GFDI**

**Figure 8: Maximum estimated Grassland Fire Danger Index (GFDI) for summer (December, January, February) of 2002—2003 (a) and 2009—2010 (b). Both zoomed areas marked with red bounding boxes for (a) and (b) are shown in (c) and (d), respectively. The fires locations for Canberra fire (c) and Toodyay fire (d) are marked with red crosshair. Forest areas are masked out in white.**

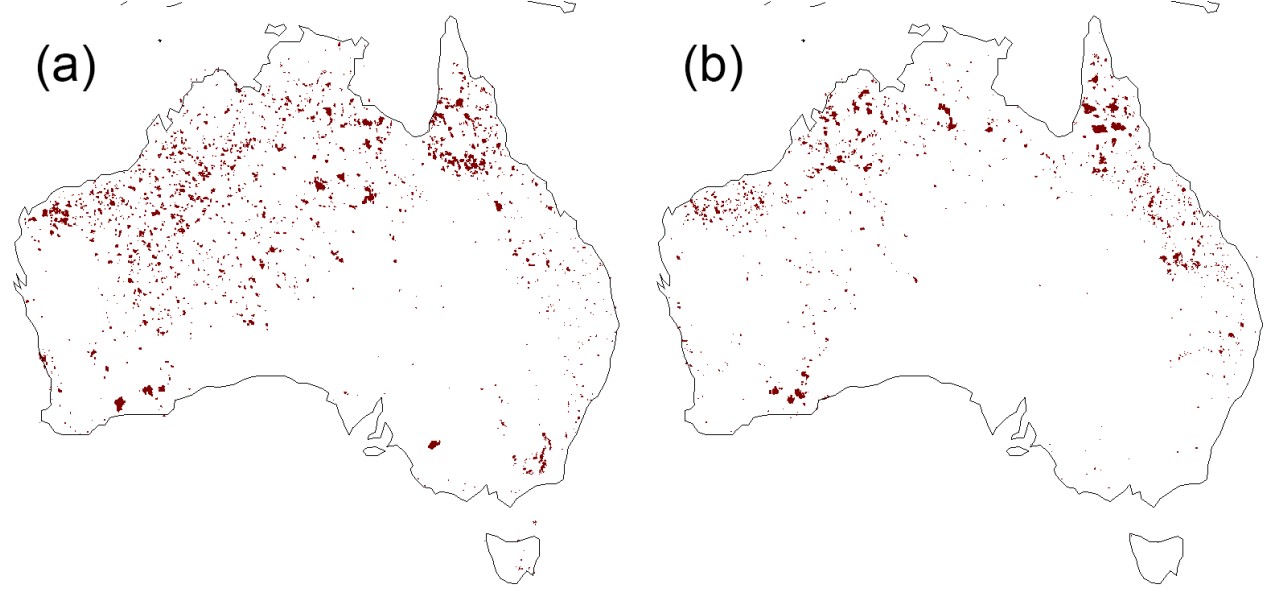

**Figure 9: MCD64A1 burned area map (Ruiz et al., 2014) during summer (December, January, February) 2002—2003 (a) and 2009—2010 (b) with forest areas masked out.**

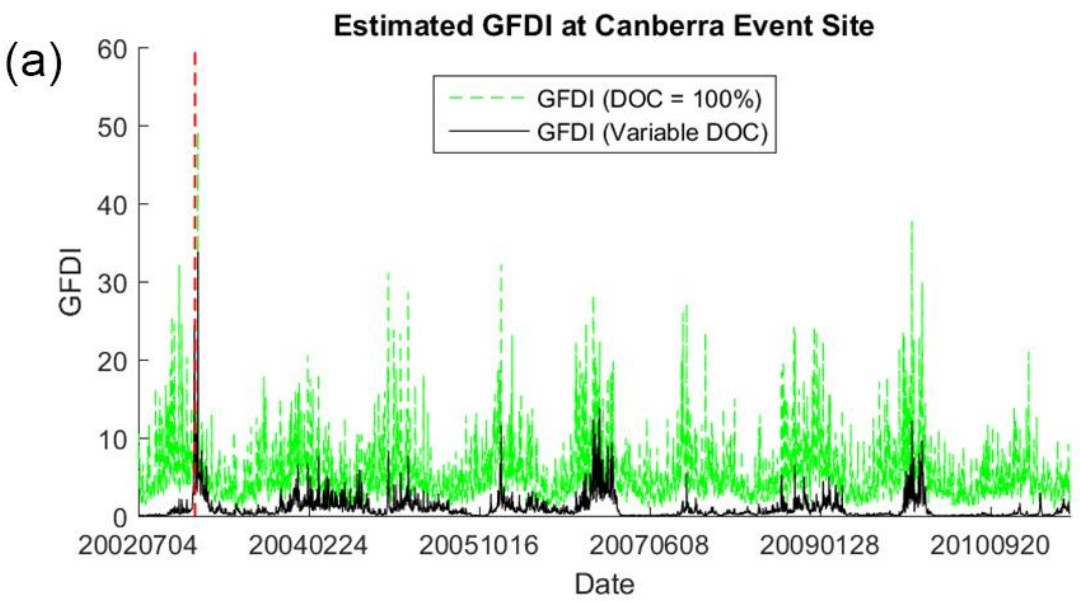

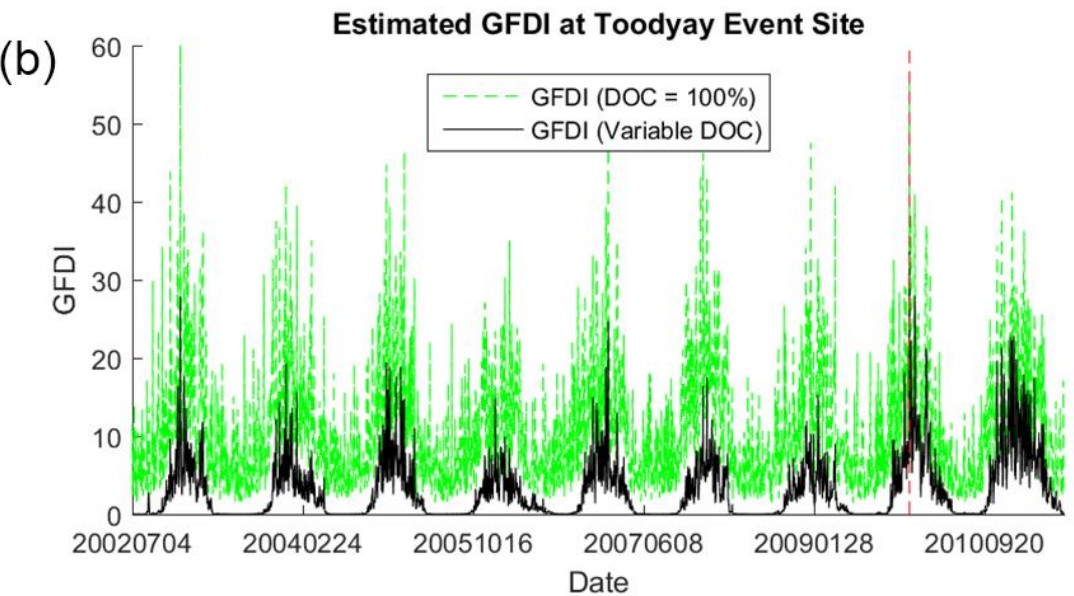

**Figure 10: Grassland Fire Danger Index (GFDI) time series plot at Weston Creek, ACT, from July 2002 to June 2011 (a) and at Toodyay, WA, from July 2002 to June 2011 (b) where the vertical dash line indicates the date of fire event on 18 January 2003 for Canberra fire and on 29 December 2009 for Toodyay fire. Solid black line is estimated GFDI time series computed from estimated degree of curing (DOC), whereas green dash line is GFDI time series computed from constant DOC at 100 %.**

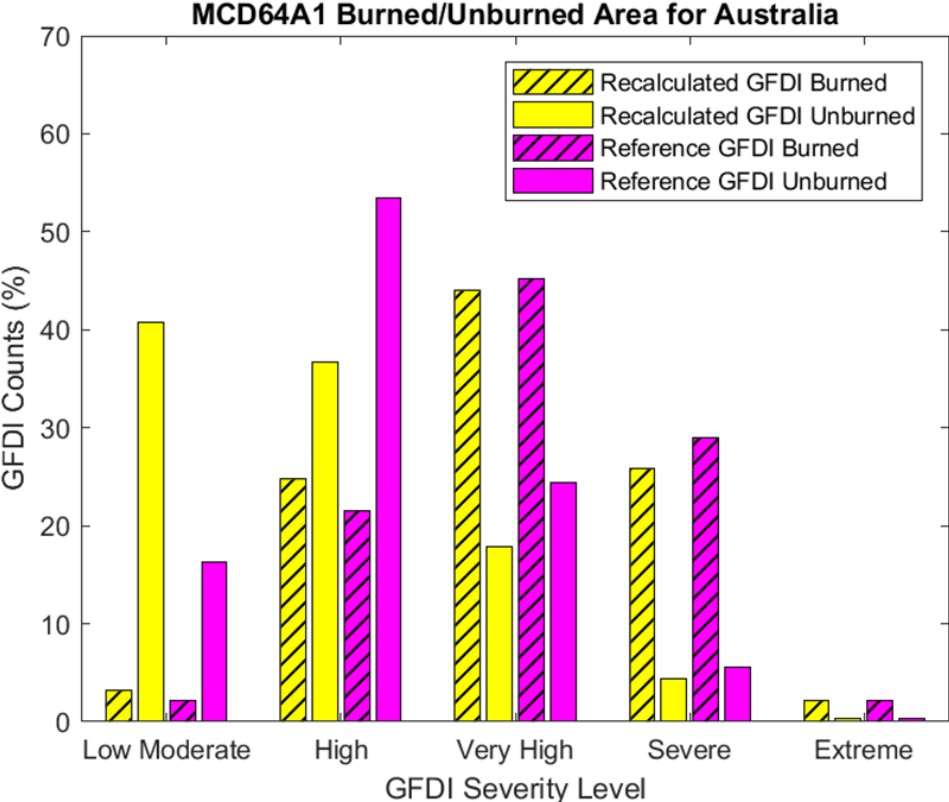

**Figure 11: Grassland Fire Danger Index (GFDI) severity level histograms at burned and unburned areas over Australia during 4 July 2002 to 26 June 2011 where the dark and light blue shaded bars are recalculated GFDI with satellite estimated variable degree of curing (DOC), while the green and yellow shaded with diagonal hatch bars are reference GFDI with constant DOC at 100 %.**

**Table 1: Calibration and evaluation of satellite based degree of curing (DOC) models derived from Vegetation Optical Depth (VOD) and Normalised Difference Vegetation Index (NDVI). Existing estimated DOC models, Method B (Newnham et al., 2010) and MapVic (Martin et al., 2015), evaluations are also listed below.**

| Model | Calibration (5/23 sites; 112/238 observations) | | All Sites Evaluation (23/23 sites; 238/238 observations) | | Independent Sites Evaluation (18/23 sites; 126/238 observations) | |
|---|---|---|---|---|---|---|
| | $r^2$ | RMSE | $r^2$ | RMSE | $r^2$ | RMSE |
| First DOC-VOD-NDVI Model [Eq. (10)] | 0.672 | 13.396 | 0.550 | 15.254 | 0.443 | 16.760 |
| Second DOC-VOD-NDVI Model [Eq. (11)] | 0.536 | 15.950 | 0.503 | 15.952 | 0.542 | 15.527 |
| Method B [Eq. (5)] | N/A | N/A | 0.611 | 14.438 | 0.632 | 11.924 |
| MapVic [Eq. (6)] | N/A | N/A | 0.435 | 19.801 | 0.562 | 14.682 |

30

**Table 2: Spatial standard deviation of estimated degree of curing (DOC) by season, month, and land cover type from 4 July 2002 to 26 June 2011.**

| DOC Spatial Standard Deviation | | | | | |
|---|---|---|---|---|---|
| **Season** | **Spatial SD (%)** | **Month** | **Spatial SD (%)** | **Land Cover Type** | **Spatial SD (%)** |
| Autumn (MAM) | 20.636 | January | 19.050 | Closed Shrublands | 11.484 |
| Winter (JJA) | 22.895 | February | 21.354 | Open Shrublands | 13.982 |
| Spring (SON) | 18.861 | March | 21.167 | Woody Savannas | 17.912 |
| Summer (DJF) | 19.161 | April | 20.228 | Savannas | 13.432 |
| | | May | 20.498 | Grasslands | 19.105 |
| | | June | 21.770 | Croplands | 20.995 |
| | | July | 23.231 | | |
| | | August | 23.634 | | |
| | | September | 21.971 | | |
| | | October | 18.305 | | |
| | | November | 16.325 | | |
| | | December | 17.276 | | |

**Table 3: Referenced and recalculated Grassland Fire Danger Index (GFDI) severity and burned—unburned area contingency table for satellite based degree of curing (DOC) derived from Vegetation Optical Depth (VOD) and Normalised Difference Vegetation Index (NDVI). Reference GFDI is computed from constant DOC at 100 %, while recalculated GFDI is computed from satellite based DOC.**

| | | Reference GFDI | | Recalculated GFDI (First Model) | |
|---|---|---|---|---|---|
| | | MCD64A1 No. of Pixels | | | |
| | | Burned | Unburned | Burned | Unburned |
| GFDI Severity | High or above | 88 | 446894217 | 80 | 319386462 |
| | Low Moderate | 5 | 395703734 | 13 | 523211489 |

| | Reference GFDI | Recalculated GFDI (First Model) |
|---|---|---|
| True Positive Rate | 0.9462 | 0.8602 |
| False Positive Rate | 0.5304 | 0.3790 |
| Accuracy | 0.4696 | 0.6210 |

30

**Table 4: Recalculated Grassland Fire Danger Index (GFDI) severity and burned—unburned area contingency table for degree of curing (DOC) computed with Method B (Newnham et al., 2010) and MapVic (Martin et al., 2015) model. Reference GFDI is computed from constant DOC at 100 %, while recalculated GFDI is computed from satellite based DOC.**

| | | Method B GFDI | | MapVic GFDI | |
|---|---|---|---|---|---|
| | | MCD64A1 No. of Pixels | | | |
| | | Burned | Unburned | Burned | Unburned |
| GFDI Severity | High or above | 9 | 131413937 | 83 | 334095499 |
| | Low Moderate | 84 | 693718749 | 10 | 488464724 |

| | Method B GFDI | MapVix GFDI |
|---|---|---|
| **True Positive Rate** | 0.0968 | 0.8925 |
| **False Positive Rate** | 0.1593 | 0.4061 |
| **Accuracy** | 0.8407 | 0.5938 |

30

**Appendix A: List of Variables**

- $C$ – degree of curing (%)
- $C_1$ – estimated degree of curing using our first model, shown in Eq. (10) (%)
- $C_2$ – estimated degree of curing using our second model, shown in Eq. (11) (%)
- $C_{Method\ B}$ – estimated degree of curing using Method B model (%) (Newnham et al., 2011)
- $C_{MapVic}$ – estimated degree of curing using MapVic model (%) (Martin et al., 2015)
- GVMI – global vegetation monitoring index (unitless)
- $f(C)$ – curing factor (unitless)
- $H_{3pm}$ – daily relative humidity at 3 pm (%)
- $NDVI_{max}$ – maximum NDVI value over a specific time range used for calculating relative greenness (unitless)
- $NDVI_{min}$ – minimum NDVI value over a specific time range used for calculating relative greenness (unitless)
- $Q$ – fuel load (kg m$^{-2}$)
- RG – relative greenness (unitless)
- $T_{max}$ – dry bulb or daily maximum temperature (° C)
- $V_{max}$ – daily maximum wind speed (km h$^{-1}$)
- $x_1$ – coefficient in the linear regression equation
- $x_2$ – coefficient in the linear regression equation
- $x_3$ – coefficient in the linear regression equation
- $x_4$ – coefficient in the linear regression equation
- $\rho_1$ – spectral reflectance measurements obtained from 620 to 670 µm region (unitless)
- $\rho_2$ – spectral reflectance measurements obtained from 841 to 876 µm region (unitless)
- $\rho_6$ – spectral reflectance measurements obtained from 1628 to 1652 µm region (unitless)

**Appendix B: List of Acronyms**

- AMSR-E (Advanced Microwave Scanning Radiometer – Earth Observing System) – one of the sensors aboard Aqua satellite; a passive microwave radiometer; stop rotating on 4 October 2011
- AMSR2 (Advance Microwave Scanning Radiometer 2) – one of the sensors aboard JAXA's GCOM-W1 satellite; a passive microwave radiometer; currently operating
- AVHRR (Advanced Very High Resolution Radiometer) – a radiation detection that is used for determining cloud cover and surface temperature
- AWAP (Australian Water Availability Project) – a state and trend of the terrestrial water balance monitoring project in Australia
- CSIRO (Commonwealth Scientific and Industrial Research Organisation) – independent Australian Federal Government's scientific research agency
- DOC (Degree of Curing) – a percentage measurement of dead material in grassland fuel complex
- ECMWF (European Centre for Medium–Range Weather Forecasts) – independent, intergovernmental organisation producing global numerical weather forecasts
- ERA–Interim – a global reanalysis climate dataset from 1979 to present
- LPRM (Land Parameter Retrieval Model) – a radiative transfer model for solving surface soil moisture and vegetation optical depth from microwave observation

- Modified Mark 4 McArthur's Grassland Fire Danger Index (GFDI) – fire danger index for grassland ecosystem developed and used in Australia
- MCD12C1 – 2010 MODIS land cover type product (University of Maryland scheme)
- MCD14ML – MODIS fire radiative power product
- MCD45A1 – daily MODIS burned area product
- MCD64A1 – daily MODIS burned area product
- MOD09A1 – 8 day MODIS reflectance product on board Terra satellite
- MODIS (Moderate Resolution Imaging Spectroradiometer) – key instrument abroad Terra and Aqua satellites acquiring data in 36 spectral bands
- NDVI (Normalised Difference Vegetation Index) – optical based satellite product used as proxy for vegetation greenness
- NCI (National Computational Infrastructure) – Australia's highly integrated and high performance research computing environment
- NOAA (National Oceanic and Atmospheric Administration) – United States government agency working on daily weather forecasts and climate monitoring
- P-Value – the significance of the results in a statistics hypothesis test where strong evidence against the null hypothesis is indicates by low p-value
- $r^2$ (R-Squared) – a statistical measurement of the correlation between two variables
- RFI (Radio Frequency Interference) – interference in microwave based satellite product (VOD in this case) due to wireless transmission, causing erroneous values in the affected grid pixels
- RMSE (Root Mean Square Error) – differences between the values estimated by a model and the observed values
- USGS (United States Geological Survey) – the United States government's scientific agency studying the landscape, natural resources, and natural hazards of the United States
- VOD (Vegetation Optical Depth) – microwave based satellite product used as proxy for vegetation moisture