# Peer review of "Estimating Grassland Curing with Remotely Sensed Data"

_Natural Hazards and Earth System Sciences, 2017_

## Referee Comment (RC1) · Anonymous Referee #1 · 19 Dec 2017

This manuscript aims to assess the performance of satellite data to improve the accuracy of the degree of grass curing (DOC) during dry summer seasons in Australia. The percentage of dead material in a grassland, as obtained by DOC, is a crucial factor for determining fire danger. The authors use satellite observed vegetation greenness (Normalised Difference Vegetation Index, NDVI) and vegetation water 15 content (Vegetation Optical Depth, VOD) information to estimate DOC. Results show a good accordance between satellite based DOC estimation and ground based observations in space and time. The authors aim also to include DOC into GFDI (Grassland Fire Danger Index) and assess if better fire severity predictions can be achieved. The overall context of the subject seems to be appropriate for this journal. Therefore, I consider that this paper could be published in Natural Hazards and Earth System Sciences

(NHESS) after the authors provide the following revisions: A. Major comments

A.1 VOD and NDVI The application of stepwise regression on Equation 4 has excluded VOD as predictor (eq. 10). Please provide in the main text a possible explanation for that feature. If NDVI is also excluded by stepwise regression is expected that the obtained model performance will decrease significantly. How much? NDVI and VOD are assumed to be (anti-) correlated. When the predictors are correlated, some of them may be insignificant in regression and their inclusion may lead to overfitting. The difference in R2 coefficient between calibration and evaluation may be also a signal of overfitting. Did the authors aware of this feature? Please consider using a cross-validation technique to evaluate model performance.

A.2 Spatial and Temporal Standard Deviation(SD) A seasonal behaviour of Spatial SD seems to be present. Further analysis of the seasons/months with higher values of SD should give additional and important information. The same analysis could be performed for each land cover type.

A.3 Spatial and Temporal Standard Deviation(SD)

Burned area maps were used as true baseline. However, the burned area map may include fires that are lit in low–moderate conditions, such as prescribed burns and fire. The less good quality of the proposed model could be associated with such type of fires that are included in low and moderate classes. Please consider using Fire Radiative Power (hotspots) as obtained by MODIS, in order to categorize burned areas according to the power (energy) released and consequently with fire intensity and severity. This will allow to eliminate low and moderate fires from your analysis and increase model accuracy, namely in case of severe fires.

B. Minor suggestions/comments

B.1 Why MODIS AQUA was not included in the analysis. The authors will have higher amount of available data and better opportunities to have valid data and to avoid clouds.
B.2 Several sites are referred by name; e.g., Darnum, Simcocks, and Neerim South, Durran Durra, Monaro, and Parry Lagoons. The authors should provide more details about the location of the sites. Non-Australian readers will get lost without those additional informations.

B.3 The sites showed in Figure 2 are the 23 sites retained from the original pool? Please clarify and introduce this information in the main text.

B.5 Page 10, line 12: Use a dot before 'With'

---

## Referee Comment (RC2) · Anonymous Referee #2 · 21 Dec 2017

GENERAL COMMENTS

This paper proposes to investigate the improvement to satellite estimation of the degree of curing (DOC) of grassland by introducing vegetation optical depth (VOD) data derived from passive microwave sensors, compared to the DOC that can be estimated using normalised difference vegetation index (NDVI) data alone. The paper then, as a second investigation, proposes to compare the reliability of the grassland fire danger index (GFDI) calculated from DOC estimated dynamically from VOD and NDVI with GFDI calculated from DOC fixed at its maximum value of 100%.

GFDI is a key parameter in operational fire management in Australia, and DOC is an essential input to the calculation of GFDI as well as being an important quantity in its own right for fire management. Thus the subject matter of this paper is certainly

important and within the scope of the journal. There has been substantial interest in recent years in using satellite data to routinely monitor DOC. Most work has used optical sensors by way of NDVI or other band combinations, and VOD offers a potentially useful complementary source of information on DOC that is more immune to gaps due to cloud and noise due to atmospheric effects.

However, the analyses that this paper presents are not done in a way that clearly demonstrates that including VOD gives better DOC estimates than using NDVI alone, let alone giving a clear indication of the extent that VOD can contribute to DOC monitoring. Also, the comparisons of GFDI calculated with different inputs do not say anything significantly new or useful. Thus this paper should not be published without a substantial revision of its analyses and other modifications. I expand on these points below.

The selection of field sites in Section 2.2 raises concerns. The rejection of sites without the expected negative correlation between VOD and curing seems very dubious: if such sites are rejected then of course the performance of a VOD-based estimate over the set of remaining sites must improve. Only five sites are accepted for the training dataset, which is a small number. Which five sites are used is not stated but should be: are they all the same grass type (improved pastures or native grasses)?; is the hummock grass site Lorna Glen included (the field data from site has very little DOC variation)?

VOD is primarily sensitive to vegetation water content (fuel moisture content), as you note, whereas NDVI and some other optical indices are sensitive to chlorophyll content. As grassland senesces the water and chlorophyll contents both decrease but not necessarily in perfect correlation, and also the relationship between DOC and FMC varies between species of grass (your reference Dilley at el. 2010). Curing is usually assessed in the field - whether by the destructive, visual or Levy rod method - by using the colour of the grass to distinguish live from dead. Since the colour is controlled by the amount of chlorophyll, a remote sensing method that responds to vegetation water would be a less direct estimate of DOC than a method that responds to chlorophyll.

Furthermore, I understand that VOD estimates the *absolute* amount of water in the vegetation and so would be controlled by the vegetation amount (biomass) as well as its fractional water content. The optical methods to estimate DOC and FMC aim to estimate *fractional* amount of live vegetation and water respectively while mitigating the confounding effects of contamination of the scene by persistent green vegetation (trees) and bare soil. Therefore the VOD method is expected to be a less direct assessment of DOC and so its use must be solidly justified. The fact that VOD is less direct suggests that perhaps a VOD anomaly approach, such as a relative VOD by analogy with relative greenness, is called for, but the authors note they found such an approach ineffective. How does VOD handle the presence of trees or bare soil in the grassland scene (pixel)?

Page 5, lines 18-21 discuss the authors' failure to reproduce Newnham et al.'s result that relative greenness (RG) predicts DOC better than plain NDVI, noting the sensitivity on the time range used to calculate RG. Newnham et al. examined the dependence on time range, but you don't cite their result or mention using it; you have not stated what time range you used. More importantly, Newnham found that the simple per-pixel RG performed worse than NDVI, as you have. Newnham only found that RG could improve on plain NDVI by using a version of RG that normalised NDVI by a spread based on climate zone, rather than per-pixel. While you were right to consider RG as a possible improved way of using NDVI to predict DOC, against which a VOD-based method should be compared, to do this properly you would have to use Newnham's preferred variant of RG. However, Newnham et al. note that the improvement is small over plain NDVI, so it may be enough just to quote Newnham's value for RMSE.

Page 5, lines 25-27 states that your search for correlations between VOD anomalies and DOC was unsuccessful. Also, page 6 line 35 to page 7 line 1 says the DOC versus VOD regression indicates that VOD alone is not reliable enough to estimate DOC. These two statements both suggest that VOD is a poor predictor of DOC, and in particular that VOD is poor at explaining the residual DOC variation beyond that

explained by NDVI. Related to this, it is surprising that Equation 10 does not include a term linear in VOD but instead VOD appears only in the cross-term (VOD)(NDVI).

It is important to acknowledge that a regression of DOC against NDVI together with any other predictor variables must mathematically give at least as good a fit as does a regression against NDVI alone. The question then becomes how well the new predictors explain the residual after the NDVI regression. From Table 1 it appears that the ratio of MODIS band 7 to band 6 (used in Method B) explains the residual better than does VOD ("Evaluation" line in Table 1). The different evaluation datasets (23 sites for the VOD method, 37 sites for Method B) make the comparison of the statistics that the authors compare for the two methods more uncertain.

Page 6, line 35 says Table 1 includes the DOC versus VOD regression results but in fact it does not. It would be instructive to see a scatter plot of DOC against VOD, or of the residual DOC unexplained by the NDVI prediction against VOD.

DOC as a function of VOD alone could be a useful alternative to optical indices in situations of prolonged cloud such as northern Australia during the monsoon. The two approaches could even be used together if they could be harmonised to be practically interchangeable, but this needs much more work.

Page 7, lines 20-35 analyse the spatial and temporal variation of DOC. It is not clear that this says anything new or that the quantitative measures of variability are useful. It is well known that DOC varies spatially and temporally, including having interannual variations. The spatial patterns and standard deviations of these are not obviously useful. What is critical is the uncertainty of estimates at any particular location and date. In any case, it is dubious to calculate these variations over the entire continent which includes non-grassland regions (e.g. forest, heath) and substantial arid or semi-arid regions for which remotely sensed characterisation of the sparse (or absent) vegetation is challenging.

The paper would be improved by including discussion of the drawbacks of VOD. For

instance, noting the magnitude of VOD errors resulting from imperfect separation of the VOD, soil surface and soil moisture contributions to the microwave radiometer signal. Also, citing any validation of the VOD dataset over Australia. A spatial resolution of 0.1 degree ($\sim$ 10 km) is fine enough for regional DOC assessments in extensive grasslands but not quite fine enough for some purposes such as input to fire behaviour models or operational GFDI calculations (currently on a 3 km or 6 km grid depending on state), or in landscapes where grassland is fragmented on small scales. You have mentioned the drawback that VOD cannot be estimated near the coast where much grassland is located.

I now turn to the analysis of GFDI with different input DOC. The conclusion that can be drawn from Figure 8 is no more than that a realistic dynamic DOC better predicts fire risk than one fixed at 100%. An NDVI-only DOC might do as well or better. It would have been more useful for you to demonstrate that GFDI is better calculated from DOC based on NDVI and VOD rather than DOC based on NDVI alone, but you have not attempted that.

Page 9, lines 20-21 notes that "the recalculated GFDI places the largest percentage of unburned pixels in the low–moderate GFDI severity class". I think that this improvement in the distribution is inevitable no matter how good or bad the DOC estimate, simply because now some fraction of pixels has DOC < 100% and so GFDI is lower for those pixels. There is no comparison for DOC estimates (< 100%) with and without VOD.

The analysis of GFDI has serious problems, which are acknowledged in Section 5.3 (e.g. forests and prescribed burns are included), that make the conclusions doubtful. Also, GFDI should be calculated from simultaneous meteorological parameters. The maximum wind speed (page 4, line 42) is often at a very different time of day and with a very different value from the 3 pm wind speed.

In light of my previous three comments, I suggest omitting the analysis and discussion of the effect of DOC on GFDI from the paper.

SPECIFIC COMMENTS

Here I list some minor points that need correction or improvement, but the list is not complete.

Page 2, line 23: You correctly note that optical based remote sensing products, including NDVI, are affected by aerosols but fail to mention that atmospheric correction can, if appropriate aerosol data is available, mitigate that. However, it could well be that VOD sidesteps this issue by being insensitive to aerosol.

Page 3, line 24: Nijs et al. (2015) is not in the reference list.

Page 4, line 10: Give the RMSE too, as well as the bias.

Page 6, line 1: There is more than one version of DOC products available, from different sources each related to the Bureau of Meteorology. State exactly how/where the data were obtained, e.g. URL of website or server.

Page 8, lines 1-2: This should also state that higher GFDI also indicates higher ignition probability (a separate factor from rate of spread of an already ignited fire).

Page 9, line 20: AVHRR data can be obtained at 1 km ($\sim$0.05°) resolution (Local Area Coverage (LAC) or High Resolution Picture Transmission (HRPT) formats).

Page 10, line 19 and Page 11, line 4: As well as NCI, it is important to acknowledge the CSIRO who produced the NCI datasets by mosaicing and regridding the tiled data provided by NASA.

Page 11, line 7: Also acknowledge the Bureau of Meteorology, who continue to generate and distribute the data products set up by the AWAP project.

Page 13, line 28: Fix spelling of "Reflectances".

Page 24, lines 18-20: Write the band wavelength ranges as, for example, "620 to 670 $\mu$m". The wavelength unit is $\mu$m, which equals 10^-6 m, not m^-6.

---

## Author Response (AR1)

**Major Comments**

**A.1** VOD and NDVI: The application of stepwise regression on Equation 4 has excluded VOD as predictor (eq. 10). Please provide in the main text a possible explanation for that feature. If NDVI is also excluded by stepwise regression is expected that the obtained model performance will decrease significantly. How much? NDVI and VOD are assumed to be (anti-) correlated. When the predictors are correlated, some of them may be insignificant in regression and their inclusion may lead to overfitting. The difference in R2 coefficient between calibration and evaluation may be also a signal of overfitting. Did the authors aware of this feature? Please consider using a cross validation technique to evaluate model performance.

Author Response: At every selected observed DOC sites (excluding the forest areas) from July 2002 to June 2011, the $r^2$ of VOD and NDVI is 0.5217 with an RMSE of 0.1117. VOD was excluded as a predictor in the final model, as expressed in Eq. (10), because during the stepwise regression, when the NDVI and (VOD)(NDVI) terms are included as the first and second predictors, the VOD term does not contribute to improving the final model prediction (i.e. p-value exceeds the acceptance threshold, preventing overfitting). When NDVI term is excluded, (VOD)(NDVI) term is included first, followed by VOD term. The summary of calibration and evaluation results with additional evaluation with independent sites only (excludes 5 sites that were used in calibration) are as listed in Table RC1-1.

Table RC1-1

| Model | Calibration (5/23 sites; 112/238 observations) | | Original Evaluation (23/23 sites; 238/238 observations) | | Total Independent Evaluation (18/23 sites; 126/238 observations) | |
|---|---|---|---|---|---|---|
| | $r^2$ | RMSE | $r^2$ | RMSE | $r^2$ | RMSE |
| C=145.565-260.817(NDVI)+137.194(VOD)(NDVI) [Eq. (10)] | 0.6724 | 13.3960 | 0.5510 | 15.2540 | 0.4430 | 16.7600 |
| C=48.699+147.603(VOD)-259.947(VOD)(NDVI) [Not shown in original paper] | 0.5355 | 15.9504 | 0.5034 | 15.9522 | 0.5423 | 15.5269 |
| C=91.637-125.219(VOD)(NDVI) [Not shown in original paper] | 0.4252 | 17.6304 | 0.3587 | 18.4265 | 0.4391 | 18.0763 |
| Method B [Eq. (5)] | N/A | N/A | 0.6110 | 14.4380 | 0.6320 | 11.9240 |
| MapVic [Eq. (6)] | N/A | N/A | 0.4350 | 19.8010 | 0.5620 | 14.6820 |

Changes in Manuscript: Additional explanations as stated in author response above has been added to section 4.1 (page 7, line 24-25, 33-38). Table 1 has been updated with the results from the model with VOD and (VOD)(NDVI) due to its better cross validation performance (shown in

Table RC1-1). The model with only (VOD)(NDVI) term is not included in the updated paper, since it is outperformed by other models. Texts and conclusions have been updated to reflect the added cross validation results, which show that Method B and MapVic outperformed DOC model with VOD (page 1, line 14-20; page 3, line 1-4; page 8, line 16-20; page 10, line 13-24; page 11, line 20-24).

**A.2** Spatial and Temporal Standard Deviation (SD): A seasonal behaviour of Spatial SD seems to be present. Further analysis of the seasons/months with higher values of SD should give additional and important information. The same analysis could be performed for each land cover type.

Author Response: Forest areas are now excluded from the analysis and Fig. 4 in the original manuscript has been replaced with Fig. RC1-1. The continental mean spatial DOC standard deviation is updated from 21.70 % to 20.39 % (page 8, line 31). Further analysis on DOC spatial standard deviation are as shown in Table RC1-2. This includes seasonal, monthly, and land cover type spatial standard deviation of DOC. From both seasonal and monthly spatial standard deviation of DOC, it is shown that DOC has the highest spatial variation during winter, which is especially true for northern Australia (Anderson et al., 2011).

[Figure]

Figure RC1-1

Table RC1-2

| DOC Spatial Standard Deviation | | | | | |
|---|---|---|---|---|---|
| **Season** | **Spatial SD (%)** | **Month** | **Spatial SD (%)** | **Land Cover Type** | **Spatial SD (%)** |
| Autumn (MAM) | 20.6355 | January | 19.0502 | Closed Shrublands | 11.4843 |
| Winter (JJA) | 22.8947 | February | 21.3538 | Open Shrublands | 13.9821 |
| Spring (SON) | 18.8605 | March | 21.1669 | Woody Savannas | 17.9117 |
| Summer (DJF) | 19.1613 | April | 20.2281 | Savannas | 13.4322 |

| | | | |
|---|---|---|---|
| May | 20.4982 | Grasslands | 19.1051 |
| June | 21.7699 | Croplands | 20.9953 |
| July | 23.2311 | | |
| August | 23.6343 | | |
| September | 21.9705 | | |
| October | 18.3045 | | |
| November | 16.3252 | | |
| December | 17.2764 | | |

Changes in Manuscript: Added the information stated in the above author response and Table RC1-2 to the results in section 4.1 (page 8; line 29-30, 33-36).

**A.3** Spatial and Temporal Standard Deviation (SD): Burned area maps were used as true baseline. However, the burned area map may include fires that are lit in low–moderate conditions, such as prescribed burns and fire. The less good quality of the proposed model could be associated with such type of fires that are included in low and moderate classes. Please consider using Fire Radiative Power (hotspots) as obtained by MODIS, in order to categorize burned areas according to the power (energy) released and consequently with fire intensity and severity. This will allow to eliminate low and moderate fires from your analysis and increase model accuracy, namely in case of severe fires.

Author Response: It is true that burned area maps are not perfect for a true baseline, since it includes prescribed burns. Following your suggestion, we use fire radiative power (FRP, unit: MW) provided in MCD14ML to mask out burned area (MCD64A1) that have low FRP. We assume any burned area with FRP lower than 100 MW to be likely prescribed burns. The changes in the GFDI and burned area analysis results (Fig. 9, Table 3, and 4 in the original manuscript) are as shown in Fig. RC1-2, Table RC1-3, and RC1-4. Note that while the true positive rate for every model significantly increases, the accuracy slightly decreases. Nevertheless, the overall results are still similar with the previous finding in the original manuscript (Method B has the highest accuracy, but worst true positive rate, MapVic has the highest true positive rate, but worst accuracy, and our proposed model sit in the middle among the three DOC models).

[Figure]

Figure RC1-2

Table RC1-3

| | | Reference GFDI | | Recalculated GFDI | |
|---|---|---|---|---|---|
| | | MCD64A1 No. of Pixels | | | |
| | | Burned | Unburned | Burned | Unburned |
| GFDI Severity | High or above | 88 | 446894217 | 80 | 319386462 |
| | Low Moderate | 5 | 395703734 | 13 | 523211489 |

| | Reference GFDI | Recalculated GFDI |
|---|---|---|
| True Positive Rate | 0.9462 | 0.8602 |
| False Positive Rate | 0.5304 | 0.3790 |
| Accuracy | 0.4696 | 0.6210 |

Table RC1-4

| | Method B GFDI | MapVic GFDI |
|---|---|---|

| | | MCD64A1 No. of Pixels | | | |
|---|---|---|---|---|---|
| | | Burned | Unburned | Burned | Unburned |
| **GFDI Severity** | **High or above** | 9 | 131413937 | 83 | 334095499 |
| | **Low Moderate** | 84 | 693718749 | 10 | 488464724 |

| | Method B GFDI | MapVix GFDI |
|---|---|---|
| **True Positive Rate** | 0.0968 | 0.8925 |
| **False Positive Rate** | 0.1593 | 0.4061 |
| **Accuracy** | 0.8407 | 0.5938 |

Changes in Manuscript: Added the description of MCD14ML to the end of section 2.1 (page 4, line 12-14). Added the above explanation in the author response regarding the application of MCD14ML on the burned area map (MCD64A1) to section 4.2 (page 9, line 26-28). Fig. 9, Table 2 and 3 will be replaced with Fig. RC1-2, Table RC1-3 and RC1-4.

**Minor Suggestions**

**B.1** Why MODIS AQUA was not included in the analysis. The authors will have higher amount of available data and better opportunities to have valid data and to avoid clouds.

Author Response: We tested both MODIS Terra and Aqua correlation with both VOD and NDVI during the initial stage of the study and found that MODIS Terra (during our study period of 4 July 2002 to 26 June 2011) has better correlation than Aqua dataset. For consistency, we decided to use only Terra dataset.

Changes in Manuscript: No changes needed.

**B.2** Several sites are referred by name; e.g., Darnum, Simcocks, and Neerim South, Durran Durra, Monaro, and Parry Lagoons. The authors should provide more details about the location of the sites. Non-Australian readers will get lost without those additional information.

Author Response: Apart from the site name, Australian states will also be provided. Grass type at each site will also be labelled for additional information.

Changes in Manuscript: All site names in the paper are now accompanied with state and grass type (page 4, line 37-38; page 5, line 2-3, 5-7). For instance, "Darnum", becomes "Darnum, VIC (mixed grass)." The name of the selected sites for calibration will also be stated in section 2.2 (page 5, line 14-15). The selected sites are: Majura, ACT (improved pasture), Tidbinbilla, ACT (mixed grass), Ballan, VIC (improved pasture), Murrayville 1, VIC (native grass), and Murrayville 2, VIC (improved pasture).

**B.3** The sites showed in Figure 2 are the 23 sites retained from the original pool? Please clarify and introduce this information in the main text.

Author Response: No, it included all 37 sites. New figure (shown as Fig. RC1-1 below) with only 23 valid sites will replace the original Fig. 2 in the paper.

[Figure]

Figure RC1-1

Changes in Manuscript: Updated Fig. 2 (shown as Fig. RC1-1 here) has replaced original Fig. 2. (page 16). Original caption will also be updated to "Figure 2: MCD12C1 land cover type map for Australia (Hansen et al., 2000). The locations of 23 valid observed degree of curing (DOC) sites are marked with crosses."

**B.4** Page 10, line 12: Use a dot before 'With'

Author Response: Thank you for pointing out the missing period.

Changes in Manuscript: Text on page 11, line 24 will be updated as noted.

**Major Comments**

**A.1** The selection of field sites in Section 2.2 raises concerns. The rejection of sites without the expected negative correlation between VOD and curing seems very dubious: if such sites are rejected then of course the performance of a VOD-based estimate over the set of remaining sites must improve. Only five sites are accepted for the training dataset, which is a small number. Which five sites are used is not stated but should be: are they all the same grass type (improved pastures or native grasses); is the hummock grass site Lorna Glen included (the field data from site has very little DOC variation)?

Author Response: While only five sites are accepted for calibration, these five have the largest number of sequential DOC records available, totalling up to 122 observations (out of 238 observations from 23 valid sites). The selected five sites are: Majura, ACT (improved pasture), Tidbinbilla, ACT (mixed grass), Ballan, VIC (improved pasture), Murrayville 1, VIC (native grass), and Murrayville 2, VIC (improved pasture).

Lorna Glen, WA (native grass - hummock) is not included in calibration, but is included in the evaluation.

A significant limitation of VOD is the large pixel size (0.1° x 0.1°). Such a large area frequently includes many land cover types and may not be dominated by grass even though a grassland observation field site falls within the pixel. The sites without the negative correlation with VOD also had no correlation with NDVI suggesting other land cover types where dominating the signal.

Changes in Manuscript: The name of the selected sites for calibration has been stated in section 2.2 (page 5, line 14-15). Total number of observations (238 DOC records) has also been added to the end of section 2.2 (page 5, line 14). Minor wording changes in page 5, line 3-4 to emphasis the reasons for site rejections due to non-negative relation between VOD and DOC.

**A.2** VOD is primarily sensitive to vegetation water content (fuel moisture content), as you note, whereas NDVI and some other optical indices are sensitive to chlorophyll content. As grassland senesces the water and chlorophyll contents both decrease but not necessarily in perfect correlation, and also the relationship between DOC and FMC varies between species of grass (your reference Dilley at el. 2010). Curing is usually assessed in the field - whether by the destructive, visual or Levy rod method - by using the colour of the grass to distinguish live from dead. Since the colour is controlled by the amount of chlorophyll, a remote sensing method that responds to vegetation water would be a less direct estimate of DOC than a method that responds to chlorophyll.

Author Response: While this is true, given the different capabilities of VOD (ability to see through cloud, smoke etc) we would like to explore the possibility of whether adding the VOD (responding to vegetation water content) will improve DOC prediction when compared with existing NDVI (responding to vegetation greenness) based DOC prediction model.

Changes in Manuscript: Added the above reasoning and reworded the study objectives (page 3, line 1-4) for clarification.

**A.3** Page 5, lines 18-21 discuss the authors' failure to reproduce Newnham et al.'s result that relative greenness (RG) predicts DOC better than plain NDVI, noting the sensitivity on the time range used to calculate RG. Newnham et al. examined the dependence on time range, but you don't cite their result or mention using it; you have not stated what time range you used. More importantly, Newnham found that the simple per-pixel RG performed worse than NDVI, as you have. Newnham only found that RG could improve on plain NDVI by using a version of RG that normalised NDVI by a spread based on climate zone, rather than per-pixel. While you were right to consider RG as a possible improved way of using NDVI to predict DOC, against which a VOD-based method should be compared, to do this properly you would have to use Newnham's preferred variant of RG. However, Newnham et al. note that the improvement is small over plain NDVI, so it may be enough just to quote Newnham's value for RMSE.

Author Response: We did not attempt to compare the spread based RG, but only range based (per pixel) RG. In Newnham et al. (2011), while range based RG performance is not as good as preferred spread based RG ($r^2 = 0.62$ and RMSE = 14.2 %), it is still better than plain NDVI (NDVI had $r^2 = 0.50$ and RMSE = 16.4 %, while 2.5 years range based has $r^2 = 0.57$ and RMSE = 15.1 %). Note that while we cannot exactly reproduce 10 years time range Newnham et al. (2011) used, since our study time frame is 9 years, we tried various 2.5 years time ranges that overlapped with Newnham et al. (2011) study period, but did not achieve the $r^2$ and RMSE as high as 0.57 and 15.1 %, respectively.

Changes in Manuscript: Cited Newnham et al. (2011) range based RG results and additional explanation (as stated in the above Author Response) to section 3.1 (page 6, line 1-6).

**A.4** Page 5, lines 25-27 states that your search for correlations between VOD anomalies and DOC was unsuccessful. Also, page 6 line 35 to page 7 line 1 says the DOC versus VOD regression indicates that VOD alone is not reliable enough to estimate DOC. These two statements both suggest that VOD is a poor predictor of DOC, and in particular that VOD is poor at explaining the residual DOC variation beyond that explained by NDVI. Related to this, it is surprising that Equation 10 does not include a term linear in VOD but instead VOD appears only in the cross-term (VOD)(NDVI).

Author Response: While it is true that VOD alone is a poor predictor of DOC, it is found that when using stepwise fit to calibrate NDVI, VOD, and (VOD)(NDVI) terms with the selected DOC

observation sites records, the addition of (VOD)(NDVI) term improves the calibration $r^2$. The VOD term does not appear as the related p-value does not meet the acceptance threshold. However, if the linear NDVI term is excluded then a linear VOD term does contribute significantly to a model with the (VOD)(NDVI) term (meeting the p-value threshold).

Changes in Manuscript: The second DOC-VOD-NDVI Model [Eq. (11)] has been added to Table 1. See all changes suggested by Referee Comment 1, RC1, A.1.

**A.5** It is important to acknowledge that a regression of DOC against NDVI together with any other predictor variables must mathematically give at least as good a fit as does a regression against NDVI alone. The question then becomes how well the new predictors explain the residual after the NDVI regression. From Table 1 it appears that the ratio of MODIS band 7 to band 6 (used in Method B) explains the residual better than does VOD ("Evaluation" line in Table 1). The different evaluation datasets (23 sites for the VOD method, 37 sites for Method B) make the comparison of the statistics that the authors compare for the two methods more uncertain.

Author Response: Method B does have better evaluation performance than the proposed model. Note that both Method B and MapVic are evaluated with the same 23 sites as our proposed model, as stated in section 3.2 (page 7, line 1-2). It should be noted that Method B used all sites in its calibration, so it is not being evaluated against any independent data, while MapVic is always being evaluated against independent data, regardless of evaluation methods as stated on page 8, lines 12-16

Changes in Manuscript: No changes needed.

**A.6** Page 6, line 35 says Table 1 includes the DOC versus VOD regression results but in fact it does not. It would be instructive to see a scatter plot of DOC against VOD, or of the residual DOC unexplained by the NDVI prediction against VOD.

Author Response: Removed left over reference to Table 1 in page 7, line 26. Thank you for pointing this out. Scatter plot for calibration of DOC against VOD, NDVI, and (VOD)(NDVI) terms is as shown in Fig. RC2-1. Another scatter plot for residual DOC unexplained by NDVI (differences between observed DOC and NDVI-based DOC) against VOD and (VOD)(NDVI) terms is as shown in Fig. RC2-2.

[Figure]

Figure RC2-1

[Figure]

Figure RC2-2

Changes in Manuscript: Changed the following sentence from "Using the linear model, as described by Eq. (3), Table 1 summarises the curing and VOD correlation result with a significant relationship and an $r^2$ of 0.20 with RMSE of 20.80 %" to "Using the linear model, as described by Eq. (3), the DOC and VOD correlation result has a significant relationship and an $r^2$ of 0.20 with RMSE of 20.80 %" (page 7, line 25-26). The scatter plots in Fig. RC2-1 and Fig. RC2-2 are also added to section 4.1 (page 7, line 26-28) as Fig. 3 and Fig. 4 (original figures from Fig. 3 onward will be shifted to Fig. 5 and so on).

**A.7** DOC as a function of VOD alone could be a useful alternative to optical indices in situations of prolonged cloud such as northern Australia during the monsoon. The two approaches could even be used together if they could be harmonised to be practically interchangeable, but this needs much more work.

Author Response: While we initially planned on predicting DOC as a function of VOD alone, we found that VOD (and its modified forms, such as VOD anomalies and seasonal based VOD) are not robust enough to be used as a lone DOC predictor.

Changes in Manuscript: No changes needed.

**A.8** Page 7, lines 20-35 analyse the spatial and temporal variation of DOC. It is not clear that this says anything new or that the quantitative measures of variability are useful. It is well known that DOC varies spatially and temporally, including having interannual variations. The spatial patterns and standard deviations of these are not obviously useful. What is critical is the uncertainty of estimates at any particular location and date. In any case, it is dubious to calculate these variations over the entire continent which includes non-grassland regions (e.g. forest, heath) and substantial arid or semi-arid regions for which remotely sensed characterisation of the sparse (or absent) vegetation is challenging.

Author Response:  The temporal and spatial variability provides an indication of areas that are likely to be more or less difficult to predict, as well as a measure to compare prediction errors to. That is, if the variability is larger than the prediction error then the prediction is likely to be useful and vice versa. Also, we are not aware of any peer reviewed literature that provides this information. Forest areas are now excluded from the analysis and Fig. 4 in the original manuscript has been replaced with Fig. RC1-1 (as shown in Referee Comments 1). The continental mean spatial DOC standard deviation is updated from 21.70 % to 20.39 % (page 8, line 31). Additional spatial variability information is added as suggested in Referee Comments 1 (RC1, A.2); see Table RC1-2 in RC1 document for more information.

Changes in Manuscript: Added the information stated in the author response of RC1, A.2 and Table RC1-2 to the results in section 4.1 (page 8, line 28-36). Fig. 4 in the original manuscript will be replaced with Fig. RC1-1.

**A.9** The paper would be improved by including discussion of the drawbacks of VOD. For instance, noting the magnitude of VOD errors resulting from imperfect separation of the VOD, soil surface and soil moisture contributions to the microwave radiometer signal. Also, citing any validation of the VOD dataset over Australia. A spatial resolution of 0.1 degree (~ 10 km) is fine enough for regional DOC assessments in extensive grasslands but not quite fine enough for some purposes such as input to fire behaviour models or operational GFDI calculations (currently on a 3 km or 6 km grid depending on state), or in landscapes where grassland is fragmented on small scales. You have mentioned the drawback that VOD cannot be estimated near the coast where much grassland is located.

Author Response: The following changes in manuscript limitation section will be added as suggested. The validation of VOD over Australia is already cited in Liu et al. (2013b, 2015) (page 2, line 40). While their work focused on the global scale, they also covered Australia.

Changes in Manuscript: Updated section 2.1 (page 3, line 19-26) with the following statements: "The VOD dataset used here is derived using the LPRM approach from which soil moisture and VOD are retrieved simultaneously. Several assumptions are made in the LPRM approach, including: canopy surface temperature equal to soil surface temperature, a constant single scattering albedo, same vegetation parameters for both Horizontal and Vertical polarizations, and minimal effect of surface roughness (Meesters et al., 2005; Owe et al., 2001). Uncertainties in soil moisture and VOD retrievals are expected with these assumptions. The evaluation of LPRM soil moisture over Australia showed that the temporal patterns of satellite-based and in situ soil moisture agree very well (Draper et al. 2009; Gevaert et al. 2016). This agreement suggests a reasonable separation of temporal patterns of soil moisture and VOD, while uncertainties may exist in the absolute magnitudes of these two variables."

Then, added the following paragraph to the beginning of section 5.3 (page 11, line 2-7): "It is worth noting here that in an operational setting atmospheric interference by clouds or smoke will cause gaps in the optical and near-infrared (NDVI) data, though the VOD data remains unaffected. We also note that while the VOD data use here was derived from the AMSR-E sensor, which is no longer operational, VOD data derived from currently operating passive microwave sensors, such as Advance Microwave Scanning Radiometer 2 (AMSR2), could be used in an operational setting. It should also be noted that VODs moderately coarse resolution of 0.1° may not be fine enough for use in many applications."

Draper, C. S., Walker, J. P., Steinle, P. J., de Jeu, R. A. M. and Holmes, T. R. H.: An evaluation of AMSR–E derived soil moisture over Australia, Remote Sensing of Environment, 113(4), 703–710, doi:10.1016/j.rse.2008.11.011, 2009.

Gevaert, A. I., Parinussa, R. M., Renzullo, L. J., van Dijk, A. I. J. M. and de Jeu, R. A. M.: Spatio-temporal evaluation of resolution enhancement for passive microwave soil moisture and vegetation optical depth, International Journal of Applied Earth Observation and Geoinformation, 45, 235–244, doi:10.1016/j.jag.2015.08.006, 2016.

**A.10** I now turn to the analysis of GFDI with different input DOC. The conclusion that can be drawn from Figure 8 is no more than that a realistic dynamic DOC better predicts fire risk than one fixed at 100%. An NDVI-only DOC might do as well or better. It would have been more useful for you to demonstrate that GFDI is better calculated from DOC based on NDVI and VOD rather than DOC based on NDVI alone, but you have not attempted that.

Author Response: While this is true, the point of Fig. 8 (currently Fig. 10) is to demonstrate that GFDI with dynamic DOC can reduce overestimation in GFDI with 100 % constant DOC. The comparison between the proposed model, Method B, and MapVic models for predicting DOC and computing recalculated GFDI (with dynamic DOC) is as stated in section 4.2 (page 9, line 35-42; page 10, line 1-7) by using burned area map.

Changes in Manuscript: No changes needed.

**A.11** Page 9, lines 20-21 notes that "the recalculated GFDI places the largest percentage of unburned pixels in the low–moderate GFDI severity class". I think that this improvement in the distribution is inevitable no matter how good or bad the DOC estimate, simply because now some fraction of pixels has DOC < 100% and so GFDI is lower for those pixels. There is no comparison for DOC estimates (< 100%) with and without VOD.

Author Response: The purpose of that specific line and Fig. 9 (currently Fig. 11) are to illustrate the different between reference GFDI (with 100% DOC) and recalculated GFDI (with dynamic DOC). The comparison between the proposed model (with VOD), Method B (without VOD), and MapVic models (without VOD) for predicting DOC and computing recalculated GFDI (with dynamic DOC) is as stated in section 4.2 (page 9, line 35-42; page 10, line 1-7) by using burned area map.

Changes in Manuscript: No changes needed.

**A.12** The analysis of GFDI has serious problems, which are acknowledged in Section 5.3 (e.g. forests and prescribed burns are included), that make the conclusions doubtful. Also, GFDI should be calculated from simultaneous meteorological parameters. The maximum wind speed (page 4, line 42) is often at a very different time of day and with a very different value from the 3 pm wind speed.

Author Response: Daily GFDI is calculated using the daily maximum wind speed (while daily relative humidity is from 3pm). While it is true that prescribed burns are included in the burned area map, we implement changes as suggested in Referee Comment 1, RC1, A.3 to minimise prescribed burns and low intensity fires. We acknowledge these problems in the section 5.3 (page 11, line 8-11).

Changes in Manuscript: Updates on burned area analysis as suggested in RC1, A3.

**A.13** In light of my previous three comments, I suggest omitting the analysis and discussion of the effect of DOC on GFDI from the paper.

Author Response: While there are some acknowledged limitations in our analysis and discussion, we feel that these elements are still important elements in the paper. That is to emphasis the different between the GFDI computed with a constant 100% DOC and dynamic, satellite based DOC whether VOD is included or not.

Changes in Manuscript: No changes needed apart from updates on burned area analysis suggested in RC1, A3.

**Specific Comments**

**B.1** Page 2, line 23: You correctly note that optical based remote sensing products, including NDVI, are affected by aerosols but fail to mention that atmospheric correction can, if appropriate aerosol data is available, mitigate that. However, it could well be that VOD sidesteps this issue by being insensitive to aerosol.

Author Response: Thank you for pointing this out. The particular line will be updated with information about atmospheric correction on optical based remote sensing product.

Changes in Manuscript: Added the following line to the end of page 2, line 25-26: "However, if appropriately detailed aerosol data is available, atmospheric correction can mitigate the aerosol effect on NDVI."

**B.2** Page 3, line 24: Nijs et al. (2015) is not in the reference list.

Author Response: Thank you for pointing this out; missing citation will be added.

Changes in Manuscript: Add the following citation to the reference list:

Nijs, A. H. A. de, Parinussa, R. M., Jeu, R. A. M. de, Schellekens, J. and Holmes, T. R. H.: A methodology to determine radio-frequency interference in AMSR2 observations, IEEE

Transactions on Geoscience and Remote Sensing, 53(9), 5148–5159, doi:10.1109/TGRS.2015.2417653, 2015.

**B.3** Page 4, line 10: Give the RMSE too, as well as the bias.

Author Response: The levy rod method DOC has RMSE of 13.5 % with a bias less than 1 % (Newnham et al., 2011).

Changes in Manuscript: Added RMSE of 13.5 % to page 4, line 24.

**B.4** Page 6, line 1: There is more than one version of DOC products available, from different sources each related to the Bureau of Meteorology. State exactly how/where the data were obtained, e.g. URL of website or server.

Author Response: The product page can be found via the following link: http://data.auscover.org.au/xwiki/bin/view/Product+pages/Grassland+Curing+MODIS+BoM

DOC datasets are downloaded from the following catalogue (Method B and MapVic): http://opendap.bom.gov.au:8080/thredds/catalog/curing_modis_500m_8-day/aust/netcdf/catalog.html

Changes in Manuscript: Added the above links to Data Availability section (page 11, line 34-37).

**B.5** Page 8, lines 1-2: This should also state that higher GFDI also indicates higher ignition probability (a separate factor from rate of spread of an already ignited fire).

Author Response: Thank you for pointing this out.

Changes in Manuscript: Add the following sentence at the end of page 9, line 10: "Though the higher GFDI does indicate higher probability of fire ignition."

**B.6** Page 9, line 20: AVHRR data can be obtained at 1 km (~0.05°) resolution (Local Area Coverage (LAC) or High Resolution Picture Transmission (HRPT) formats).

Author Response: Thank you for pointing this out.

Changes in Manuscript: Removed the following statement from page 10, line 29: "…the advantage of high resolution offered by MODIS (0.05° for AVHRR, but 0.005° for MODIS), or…"

**B.7** Page 10, line 19 and Page 11, line 4: As well as NCI, it is important to acknowledge the CSIRO who produced the NCI datasets by mosaicing and regridding the tiled data provided by NASA.

Author Response: Thank you for pointing this out.

Changes in Manuscript: Added acknowledgement to CSIRO in page 11, line 31 and page 12, line 17.

**B.8** Page 11, line 7: Also acknowledge the Bureau of Meteorology, who continue to generate and distribute the data products set up by the AWAP project.

Author Response: Thank you for pointing this out.

Changes in Manuscript: Added acknowledgement to the Bureau of Meteorology in page 12, line 18-19.

**B.9** Page 13, line 28: Fix spelling of "Reflectances".

Author Response: Thank you for pointing this out.

Changes in Manuscript: Fix the spelling as pointed out in page 15, line 12.

**B.10** Page 24, lines 18-20: Write the band wavelength ranges as, for example, "620 to 670 µm". The wavelength unit is µm, which equals 10ˆ-6 m, not mˆ-6.

Author Response: Thank you for pointing this out.

Changes in Manuscript: Correct m$^{-6}$ to µm (page 27, line 20-22).

**Estimating Grassland Curing with Remotely Sensed Data**

Wasin Chaivaranont[1], Jason P. Evans[1], Yi Y. Liu[1, 2], Jason J. Sharples[3]

[1]ARC Centre of Excellence for Climate Systems Science and Climate Change Research Centre, UNSW, Sydney, 2052, Australia

[2]School of Geography and Remote Sensing, Nanjing University of Information Science and Technology, Nanjing, 210044, China

[3]School of Physical, Environmental and Mathematical Sciences, UNSW, Canberra, ACT 2600

*Correspondence to:* Wasin Chaivaranont (w.chaivaranont@student.unsw.edu.au)

**Abstract.** Wildfire can become a catastrophic natural hazard, especially during dry summer seasons in Australia. Severity is influenced by various meteorological, geographical, and fuel characteristics. Modified Mark 4 McArthur's Grassland Fire Danger Index (GFDI) is a commonly used approach to determine the fire danger level in grassland ecosystems. The degree of curing (DOC, i.e. proportion of dead material) of the grass is one key ingredient in determining the fire danger. It is difficult to collect accurate DOC information in the field, therefore, ground observed measurements are rather limited. In this study, we explore the possibility of whether adding the satellite observed data responding to vegetation water content (Vegetation Optical Depth, VOD) will improve DOC prediction when compared with the existing satellite observed data responding to vegetation greenness (Normalised Difference Vegetation Index, NDVI) based DOC prediction models. we used satellite observed vegetation greenness (Normalised Difference Vegetation Index, NDVI) and vegetation water content (Vegetation Optical Depth, VOD) information to improve the accuracy of the DOC estimation. First, a statistically significant relationships is are established between selected ground observed DOC and satellite observed vegetation datasets (NDVI and VOD) with an $r^2$ of up to 0.67. DOC levels estimated using satellite observations were then evaluated using field measurements with an $r^2$ of 0.44 to 0.55. Results suggest that satellite VOD based DOC estimation can reasonably reproduce ground based observations in space and time and is comparable to the existing NDVI based DOC estimation models. Comparison with currently available satellite based DOC products shows that our model has a comparable and arguably more balanced performance.

**Copyright statement**

The article is distributed under the Creative Commons Attribution 3.0 License. Unless otherwise stated, associated published material is distributed under the same licence.

[revised manuscript text omitted]

---

## Author Response (AR2)

**nhess-2017-101-RC2 Author Comments**

**Specific Comments**

**B.1** It would be good if you can quote in your paper any relevant numbers from the Liu et al. references on the magnitude of VOD errors relevant to grassland in Australia.

Author Response: While there are no numbers regarding directly relevant magnitude of VOD errors relevant to grassland in Australia from past Lie et al. papers, a recent paper by Chen et al., 2018 (as cited below) mentioned a very high correlation between VOD and aboveground biomass in Australian tropical savannahs that includes large area of northern grassland with an $r^2$ of 0.96.

Chen, X., Liu, Y. Y., Evans, J. P., Parinussa, R. M., Dijk, A. I. J. M. van and Yebra, M.: Estimating fire severity and carbon emissions over Australian tropical savannahs based on passive microwave satellite observations, Int. J. Remote Sens., doi:10.1080/01431161.2018.1460507, 2018.

Changes in Manuscript: Add the following statement to page 2, line 41 to 42: "A recent study on estimating fire severity based on VOD changes also demonstrated that VOD has high correlation with aboveground biomass in Australian tropical savannahs that includes large area of northern grassland with an r2 of 0.96 (Chen et al., 2018)." The citation for Chen et al., 2018 is also added to the references (page 13, line 4 to 6).

**B.2** Page 4, line 28 and line 36: Change "land use" to "land cover". They are different.

Author Response: Agreed.

Changes in Manuscript: Corrections are made as suggested.

**B.3** Page 5: In line 37 append "s" to "alternative, and in line 38 change "purposed" to "proposed".

Author Response: Agreed.

Changes in Manuscript: Corrections are made as suggested.

**B.4** New Figure RC1-2: The use of colour is more confusing than it needs to be: for four classes you have used four colours plus the presence or absence of hatching. Use two colours only, and with and without hatching to give the four possibilities needed. To me the dark hatching suggests "burned", so I suggest using the two colours to distinguish Referenced from Recalculated.

Author Response: Agreed; both recalculated and reference GFDI each have their corresponding colour and the presence of hatch indicates burned area.

Changes in Manuscript: Replace Figure 11 (or Figure RC1-2 from the previous author comments) on page 23 with the new colours and hatch scheme as suggested. Replaced figure is also as shown below.

[Figure]

Figure 11 (or Figure RC1-2 from the previous author comments)

**B.5** Page 7, line 24: The grammar is incorrect and the meaning is unclear for "At every selected observed DOC sites (excluding the forest areas)". Does it mean "Across all selected observed DOC sites (excluding the forest areas)"?

Author Response: Yes, it is incorrect.

Changes in Manuscript: Correction is made as suggested.

**B.6** Page 7, sentence on lines 36-38: Insert "the" before each of "NDVI", "(VOD)(NDVI)" and "VOD".

Author Response: Agreed.

Changes in Manuscript: Corrections are made as suggested.

**B.7** Page 9, line 10: The grammar is wrong for the sentence starting "Though …". For better wording, I suggest deleting this sentence and inserting "higher probability of fire ignition and" after "indicates" in line 8.

Author Response: Yes, that is not a correct grammar.

Changes in Manuscript: Corrections are made as suggested.

**B.8** Page 10, line 19: Delete "models".

Author Response: Agreed.

Changes in Manuscript: Correction is made as suggested.

**B.9** Page 11, line 3: Delete "and near-infrared", since optical includes infrared. In the same line, change "use" to "used".

Author Response: Agreed.

Changes in Manuscript: Corrections are made as suggested.

**B.10** Page 11, line 5: Change "as Advance" to "as the Advanced".

Author Response: Agreed.

Changes in Manuscript: Correction is made as suggested.

**B.11** Page 11, line 6: Change "VOD" to "VOD's".

Author Response: Agreed.

Changes in Manuscript: Correction is made as suggested.

**B.12** Page 11, line 11: Change "active" to "the MODIS active".

Author Response: Agreed.

Changes in Manuscript: Correction is made as suggested.

**B.13** Page 11, line 15: I think "500 m$^2$" is wrong. That area corresponds to a square of approximate dimensions 22 m x 22 m. If you mean the size of a pixel of the MODIS DOC product, then write it unambiguously as "500 m x 500 m" or just "500 m".

Author Response: Thank you for the suggestion.

Changes in Manuscript: Change from 500 m$^2$ to 500 m, as suggested.

**B.14** Page 11, line 24: Change "detecting" to "predicting".

Author Response: Agreed.

Changes in Manuscript: Correction is made as suggested.

**B.15** Page 27, line 27: Change "radiometre" to "radiometer".

Author Response: Agreed.

Changes in Manuscript: Corrections are made as suggested (also on page 27, line 25, as well).

**B.16** Page 27, line 28: Change "detection that" to "detector that is".

Author Response: Agreed.

Changes in Manuscript: Correction is made as suggested.

**B.17** Page 28, line 16: Change "significant" to "significance", if that is correct.

Author Response: Yes, it should be significance (noun), rather than significant (adjective).

Changes in Manuscript: Correction is made as suggested.

**B.18** Page 28, line 18: Change "closeness of the data to the fitted regression line" to correlation between two variables. The value of $r^2$ is affected not only by the spread about the regression line but also the range of values spanned.

Author Response: Agreed.

Changes in Manuscript: Correction is made as suggested.